# Lost in the North Sea—Geophysical and geoarchaeological prospection of the Rungholt medieval dyke system (North Frisia, Germany)

**Dennis Wilken**[1]*, **Hanna Hadler**[2], **Tina Wunderlich**[1], **Bente Majchczack**[3], **Michaela Schwardt**[1], **Annika Fediuk**[1], **Peter Fischer**[2], **Timo Willershäuser**[2], **Stefanie Klooß**[4], **Andreas Vött**[2], **Wolfgang Rabbel**[2]

**1** Institute of Geosciences, Christian-Albrechts-University, Kiel, Germany, **2** Institute for Geography, Johannes Gutenberg-University Mainz, Mainz, Germany, **3** Cluster of Excellence ROOTS, Christian-Albrechts-University, Kiel, Germany, **4** Archaeological State Department Schleswig-Holstein, Schleswig, Germany

* dennis.wilken@ifg.uni-kiel.de

**Data Availability Statement:** All core stratigraphies are made available to the Schleswig-Holstein State Geological archives. Archaeological

## Abstract

We performed geophysical and geoarchaeological investigations in the Wadden Sea off North Frisia (Schleswig-Holstein, Germany) to map the remains and to determine the state of preservation of the medieval settlement of Rungholt, especially its southern dyke segment, called the *Niedam* dyke. Based on archaeological finds and historical maps, Rungholt is assumed to be located in the wadden sea area around the island Hallig Südfall. During medieval and early modern times, extreme storm events caused major land losses, turning cultivated marshland into tidal flats. Especially the 1st Grote Mandrenke (or St. Marcellus' flood), an extreme storm surge event in 1362 AD, is addressed as the major event that flooded and destroyed most of the Rungholt cultural landscape. Cultural traces like remains of dykes, drainage ditches, tidal gates, dwelling mounds or even plough marks were randomly surveyed and mapped in the tidal flats by several authors at the beginning of the 20th century. Due to the tidal flat dynamics with frequently shifting tidal creeks and sand bars, the distribution of cultural remains visible at the surface is rapidly changing, making it hard to create a comprehensive map of the cultural landscape by surveying. Today, the Niedam dyke area is fully covered by tidal flat sediments, depriving any remains from further archaeological investigation. Since little is known about the precise location or state of preservation of these remains, our investigation aimed at the rediscovery of the medieval dyke system and associated structure with modern and accurate geophysical, geodetical and geoarchaeological methods. Magnetic gradiometry revealed a large part of the medieval dyke, confirming two tidal gates and several terps connected inland with the dyke, providing a detailed example of a Frisian medieval dyke system. Based on our results, the so far inaccurate and incomplete maps of this part of Rungholt can now be specified and completed. Beyond that, seismic reflection profiles give a first depth resolving insight in the remains of the dyke system, revealing a severe threat to the medieval remains by erosion. The site is exemplary for

find data are recorded by the State Archaeological office and integrated into the Schleswig-Holstein Archaeological Database (ADSH). Research data are entered into the web-based information system of the cultural landscape cadastre KuLaDig (www. KuLaDig.de, Ickerodt 2017) introduced in Schleswig-Holstein and coordinated by the Rhineland Regional Council and made available to a broad public. All basic data as well as new results are entered into a joint GIS project that is accessible via a central project server of Johannes-Guttenberg-University Mainz in cooperation with the Center for Data Processing at JGU.

**Funding:** This study was carried out within the framework of the priority program 1630 -Harbours from the Roman Periods to the Middle Ages- by the German Research Foundation (RA 496/26-2, ZI 721/10-2, 20/7-2, HA 7647/1-1, VO 938/21-1) and the ROOTS Cluster of Excellence funded by the German Research Foundation (EXC 2150-390870439). We also acknowledge financial support by the German Research Foundation within the funding programme Open Access Publikationskosten for assisting with the costs of the publication fees. The funders had no role in study design, data collection and analysis, decision to publish, or preparation of the manuscript.

**Competing interests:** The authors have declared that no competing interests exist.

the entire North Frisian coast, that was influenced by multiple flood events in the middle ages to modern times.

## Introduction

The Wadden Sea region stretches between the north-western Netherlands and southern Jutland along the North Sea coast, being an area of dyked or formerly dyked salt marshes and reclaimed coastal peat bogs [1]. This coastal wetland contains visible and past human adaptations to the environment in the form of embankments, dykes, canals, polders, making it a cultural landscape of exceptional cultural historical value with a strong maritime character [1, 2]. The region has a complex settlement history, with first long-term habitations in the salt marsh areas starting in the Iron Age around 600 BC (summarized by [2], 116-122). Natural and human-influenced dynamics have changed the marshes and tidal flats throughout time. In the study area, the tidal flats of North Frisia (Germany), these changes are especially visible by the numerous traces of medieval and early modern settlements and remains of their cultural landscapes which were ultimately lost to the sea, appearing and disappearing in the ever-changing environment of the Wadden Sea. In the Rungholt area, a medieval settlement of historical record, most archaeological remains date to the 12th to 14th cent. AD, a period when a wave of immigrating Frisian settlers extensively cultivated the wide coastal marshes and fenlands. Today, only little is known about the appearance of this medieval landscape. During the Medieval and Early Modern Periods, major storm events in 1362, 1634 and 1717 significantly reshaped the North Frisian coastal area, destroying wide parts of the cultural landscape and turning them into the tidal flats of the present-day Wadden Sea. During the 14th century and possibly mainly triggered by the historical 1362 AD event, the coastline in the study area retreated about 25 km inland, with only minor patches of marshland left in between [3]. The part of Rungholt, that is investigated in the present study, (Fig 1) is well known for its high density of cultural remains: groups of dwelling mounds, a wide network of drainage ditches, large dykes and the remains of two wooden tidal gates [6, 7] were first observed by [8, 9], when areas of medieval marshland were exposed after erosion of parts of the overlying, geologically younger tidal island of Südfall [10]. These remains are assumed to be a central part of the Rungholt dyke system. Rungholt was known from historical tradition to have been lost in the flood of 1362 AD (e.g. [8, 11–14]). Archaeological finds from the tidal flats around Hallig Südfall indicate the importance of the medieval settlement. They include imported goods from the Rhineland, Flanders and even Spain, namely pottery, metal vessels, metal ornaments and weapons [15]. West and southwest of Südfall, two distinctive settlement areas can be distinguished: the so called *8-Warften area* and the *Niedam area*, whereas the latter was decribed by ([8, 16, 17]) as a possible harbour site, comprising features typical of a tidal-gate associated harbour (so-called "Sielhafen", [18]). Fig 2a maps the significant structures that were visible at the surface at the beginning of the 20th century as described by [8, 17, 19]. Fig 2b sketches the features described in terms of the dyke systems. In the western part of the investigated dyke section, remains of two wooden tidal gates were described—dating to the 12th to 14th cent. AD [20]. The inner chambers have a length of ca. 25 m, indicating a dyke base of 25 m to 30 m. So far, contemporary gates of similar size are only known from the medieval harbour of Rotterdam [21]. On the landward side, rectangular dwelling mounds are directly attached to the dyke and provide sufficient space for settlement and trading activites (i.e. houses, storage facilities). Detached dwelling mounds mainly belong to farmsteads.

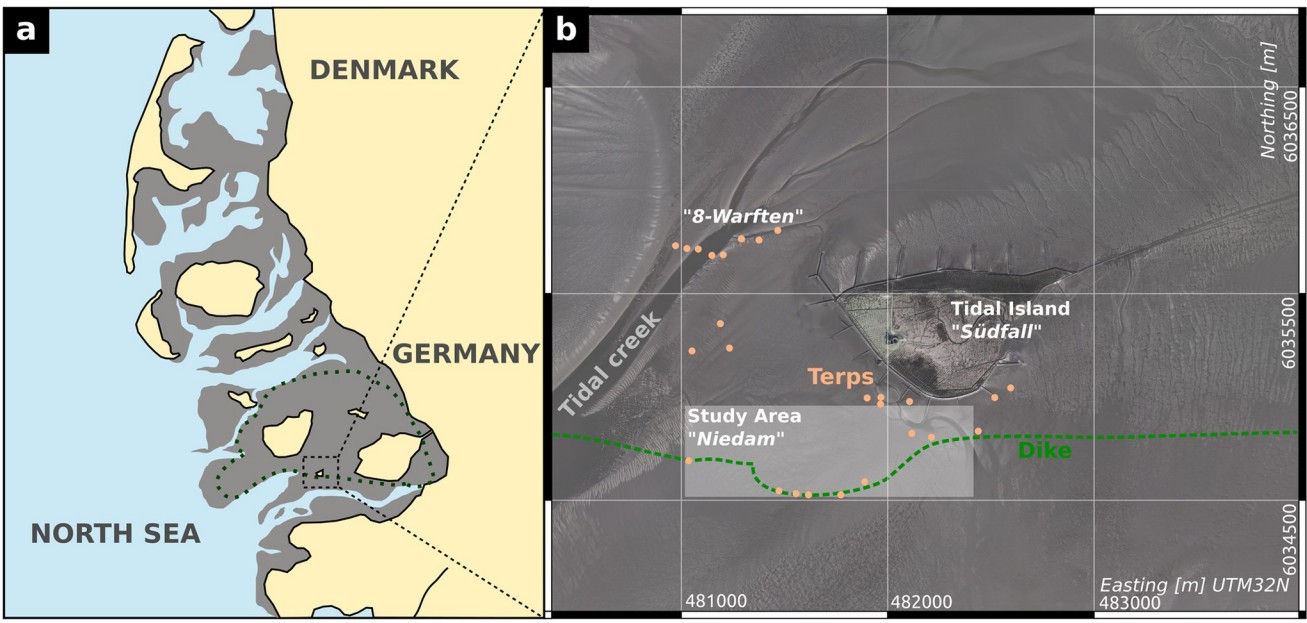

**Fig 1. Study site map.** a) Location of the study area at the North Sea coast of Germany. Grey areas show the tidal flats and yellow areas show the recent coastline, redrawn after [4]. The dashed line indicates an approximate model of the former coastline of the island *Strand* before the Marcellus flood 1362 AD. b) Aerial photo of the tidal island *Hallig Südfall* and surrounding tidal flats [5]. The white transparent box indicates the investigated site, named *Niedam area*, dots mark known archaeological findings recorded in the catalogue of the archaeological state department of Schleswig-Holstein. Aerial photos are a combination of photos 32480_6034, 32480_6036, 32482_6034, and 32482_6036, republished from Regional government authority for coastal preservation, national park and ocean protection (LKN) under a CC BY license, with permission from LKN, original copyright 2013.

With the presented geoarchaeological and geophysical prospection approach we aim first and foremost at determining the exact location and extent of the cultural remains connected to the dyke system in the Niedam area, as they are described and mapped in the older research tradition (e.g. [17]). By using modern prospection techniques including high resolution depth imaging methods, we aim at a comprehensive picture of the archaeological features; advancing from repeated, random surveying and sighting in the past towards a systematic prospection. Based on the prospection data, we seek to re-interpret the Niedam-dyke settlement and harbour area. Furthermore, we will explore to which extent the archaeological structures are preserved after several decades of erosion, flooding events and ongoing sediment transport due to the ever-changing tidal flat morphology since their original discovery in the 1920s. This leads to the question, to which degree the archaeological features beneath the tidal flats have been eroded and how pressing the need for further exploration in the tidal flats is.

## History of land use and reclamation in the Rungholt area

The preconditions for medieval settlement dynamics largely depended on the palaeogeographic evolution of the North Frisian coastal area during the Holocene. In the course of the post-glacial sea level rise, sand-spits formed west of today's Wadden Sea, closing the area off from a direct marine influence and leading to the formation of a quiescent brackish lagoonal system repeatedly affected by extensive peat formation ([10, 22]). Since 2000 BC, ongoing siltation has formed wide fenlands and marshes, traversed by different types of waterbodies. In the mid-1st millennium BC, the marine influence is almost negligible [23]. The first known usage of the fenlands was by man in the late Neolithic and early Bronze Age ([10, 24]). First settlements sites are indicated by few finds of the Roman Iron Age which hint at a thin and short-

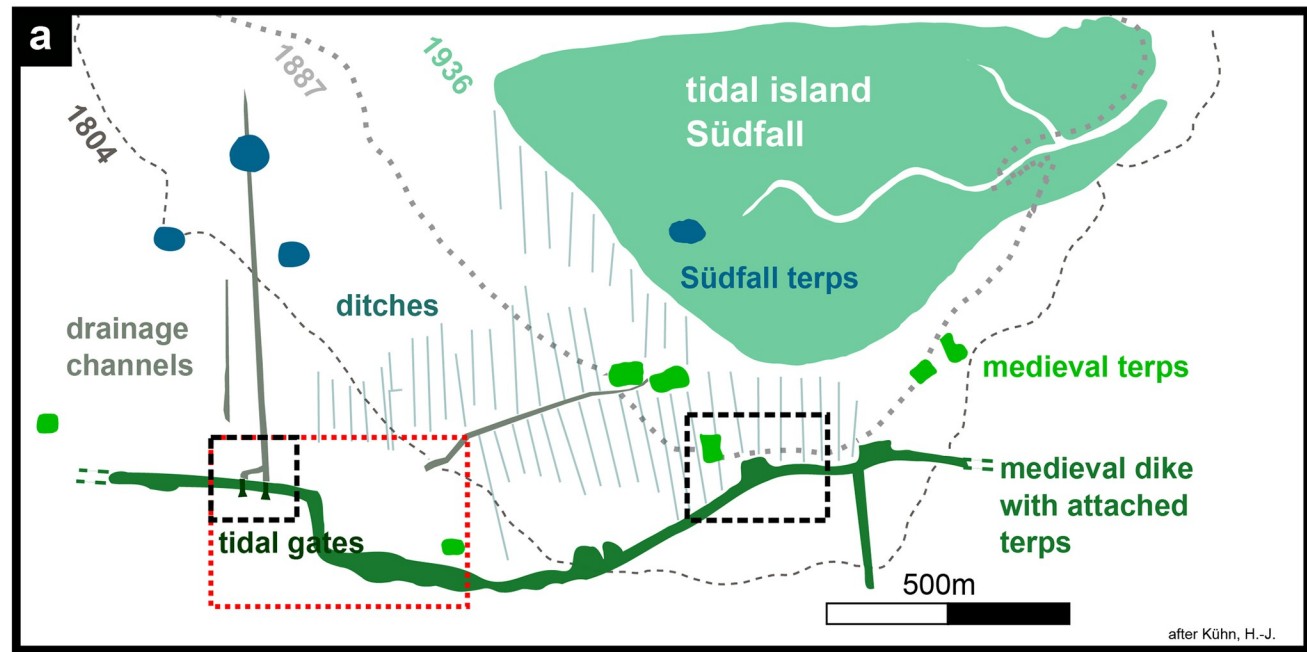

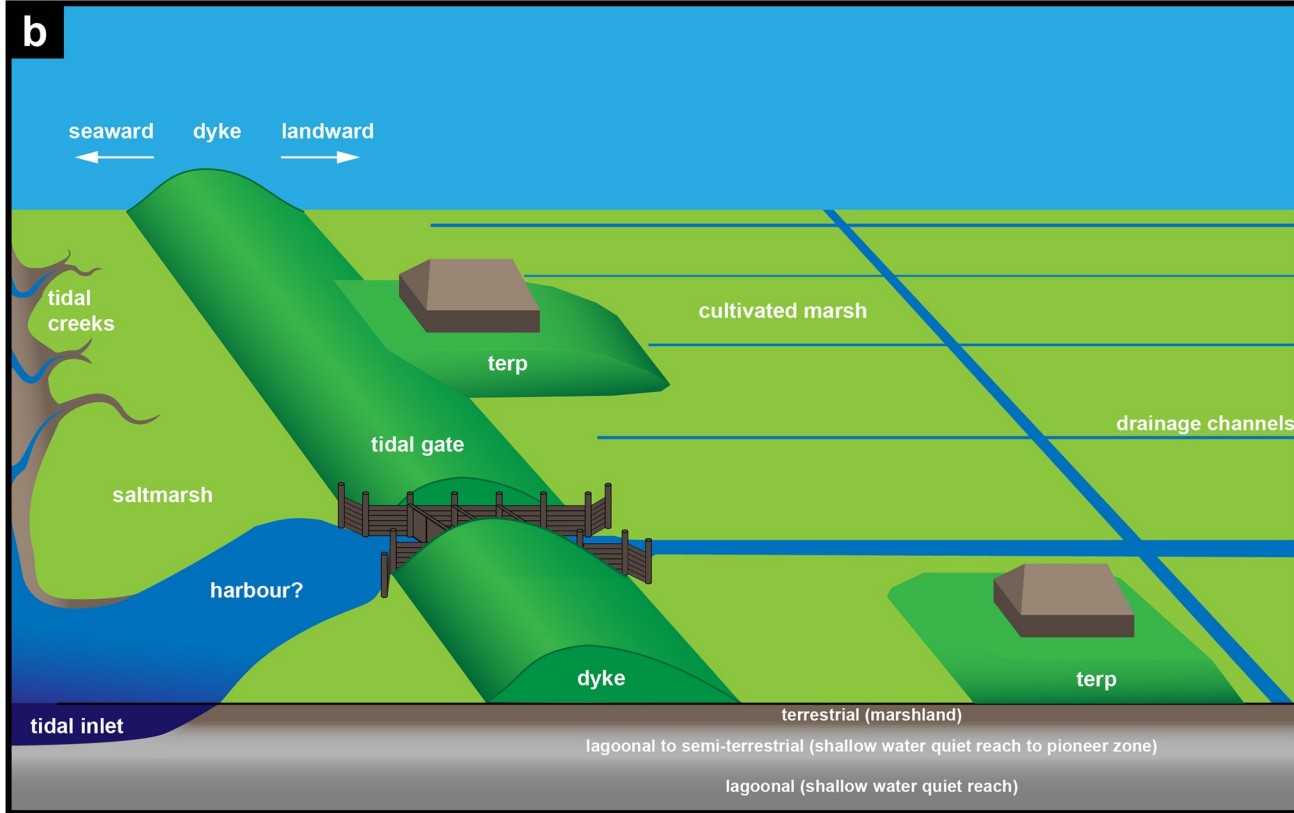

**Fig 2. Findings and dyke model.** a) map summarizing all significant archaeological structures that were visible in the tidal flats and recorded by a local resident during the early 20th century after [8, 17]. Drawn maps of these sources were georeferenced, combined and redrawn for this figure. Observations were usually located by bearing a fix object on the mainland or the island and triangulation [17]. b) sketch illustrating the main features of the coastal environment connected to the medieval dyke. Typical rectangular dwelling mounds were either attached to the dyke or lay isolated in drained cultivated marshlands and protected the settlements from being flooded. The report by [8] and [9] further shows two tidal gates embedded in the dyke, that opened during low tide to drain the marshes on the landward side but closed with the incoming tide.

lived settlement horizon. It was not before the 8[th] cent. AD that a first group of Frisian immigrants settled in favorable, elevated coastal marshes on unprotected level-ground settlements [6] during a time of stagnation in sea-level rise in the 8[th]-10[th] century ([25, 26]). It appears, that from the 11[th] century AD onward, an increased marine influence endangered the low marshlands of North Frisia by flooding events ([10, 22, 23, 27, 28]). Immediate reactions to protect settlements from flooding were the construction of artificial dwelling mounds or terps. At the same time, the settlers sought to reclaim larger areas of marsh- and fenlands for agriculture, which had to be protected from floods. Thus, first dykes, constructed in the late 11[th] century AD, were either built as ring dykes around farmland or between dwelling mounds, enclosing land for farming and settlements ([2, 6, 29, 30]) and forming polders (= koog or groden). Archaeological observations on high medieval dykes and dwelling mounds showed that contemporary dykes reached only heights of up to 1.6 m to 2.0 m above sea-level. Therefore, it has been argued that the primary function of the early dykes was to protect the farmland against seasonal flooding during the summer and only the terps were sufficiently protected against major storm surges ([11, 27, 29]). While the early dyking efforts were confined to small and dispersed areas, large-scale land reclamation of the 12[th]-13[th] century required construction of long coastal dykes to enclose and protect large areas. Consequently, the dyked inland had to be drained through tidal gates that could at the same time serve as landing sites or harbours ([7, 18]). The know-how needed for dyke construction, fenland-cultivation and water-management was probably brought by a second wave of Frisian immigrants from the southern North Sea coast ([6, 29, 31]). Peat extraction for soil melioration, firing and salt production as well as intense draining of marshland caused subsidence of the ground surface, increasing the potential for flooding and permanent land losses after breaching of dykes as it is recorded for the major flooding events in 1362 AD and later ([20, 22, 32]).

## Methods

The tidal flat environment of the study area (Fig 1b bears several difficulties for performing geophysical and geoarchaeological prospection. Several geophysical techniques are not feasible, yet the saline character of the intertidal zone is a restricting factor [33]. For example, ground penetrating radar has a severely reduced depth penetration in saline/brackish wetland environments because of signal damping (e.g. [34]). There are also several practical limitations. First, there is a very narrow time window whilst the area is accessible by walking during low tide. This means, that slow progressing methods like electrical resistivity tomography or land based seismic methods are not feasible for prospecting large areas. The second limitation is the general accessibility. The main investigation area lies about one kilometer off Hallig Südfall. This distance has to be considered and passed during the low tide time window, leaving about two hours of measuring time for one low tide phase. Furthermore, accommodation is neither available nor allowed on the Hallig, making the prospection team dependant on daily transfer from the mainland to the island (about 7 km). During high tide, boat based measurements can be performed, starting at a jetty in the south of Südfall. Measurement time is slightly longer than two hours, and is only reduced by the water depth (about 1 m) at which the boat is able to leave and return to the island. This leads to the following available datasets in the tidal flat area around the southern part of the Rungholt dyke system, that were collected in two field campaigns in 2016 and 2017 in the framework of the DFG Priority Programme SPP 1630: 'Harbours from the Roman Period to the Middle Ages' [35]:

1. **Magnetic gradiometry** data, collected two hours each day during good weather conditions and good sight. The data can be acquired on tidal flat areas that fall dry during low tide. The

area that is accessible by walking with a sensor cart is mainly made of sandbars with only a small portion of mud.

2. **Marine reflection seismic** data, collected about two to three hours a day during high tide and calm sea conditions. Wave height should be less than half a meter to avoid wave motion noise and image distortion due to tilted source and receivers.

3. **Percussion core samples** and subsequent laboratory analysis during low tide.

4. Some archaeological features become visible time by time due to erosion of the tidal flats. Thus archived **aerial photos** can show archaeological features that are no more visible today, thereby improving parts of the model in Fig 2a.

## Magnetic gradiometry

Magnetic gradiometry measures the vertical difference of the vertical component of the magnetic field of the Earth. The difference is independent of regional fields and therefore mainly measures the local magnetic field caused by bodies in the shallow part of the ground. A magnetic gradiometer survey was performed using an array of six fluxgate gradiometers (Foerster fluxgate differential vertical component magnetometers) with an internal vertical sensor distance of 0.65 m, a horizontal sensor spacing of 0.5 m and a sampling frequency of 20 Hz mounted on a cart especially built for the tidal flat environment (Fig 3a). Accurate positioning was achieved by RTK (Real Time Kinematic) DGNSS (Leica 530). Data processing was performed using our own software package in Matlab. The arithmetic mean was subtracted from the data of each profile in order to eliminate the constant portion of the magnetic field caused by instruments installed on the sensor cart. Noisy profiles are removed and some profiles are high pass filtered with a cutoff of $k = 0.002$ 1/sample if they still showed a long wavelength trend after mean subtraction. All remaining profiles are low-pass filtered with a cutoff wavenumber of $k = 0.06$ 1/sample to remove short wavelength noise, such as walking or movement noise. The resulting values were binned inline and interpolated cross-line (using linear interpolation of Matlab's griddata function) between neighbouring profiles in order to form a data grid of 0.2 m bin size. Finally the area is kk-filtered using a pieslice filter operator of 20° opening, cosine tapered at the edges, to remove remaining stripe noise from the data, following [36].

## Marine reflection seismics

We used a high resolution two channel seismic reflection (sediment echosounder) system, and RTK-DGNSS positioning mounted on an inflatable catamaran (Fig 3b). Comparable prospection setups have been used by e.g. [37–40]. The seismic acquisition system consisted of a piezoelectric transducer (ELAC Nautik TL-444, 4 kHz center frequency) that acts as seismic source, and two hydrophones. The transducer is driven with a Fuchs-Müller wavelet of 4 kHz center frequency. After passing the transfer functions of source and hydrophone transducer, the system creates a signal with a bandwidth from 2 kHz to 6 kHz and a peak frequency of 3.5 kHz. Taking 3.5 kHz as resulting peak frequency and 1480 m/s as wave velocity in water, the theoretical vertical resolution is about 0.1 m. The horizontal resolution is defined by the first Fresnel zone, which depends on the depth of the reflector and ranges from 0.45 m (0.5 m depth) to 3.5 m (30 m depth). However, as we were not aiming at imaging small scale objects but at imaging the stratigraphy of mainly 2D structures (dyke) or structures with a horizontal extent of several meters (terps), the horizontal resolution is not as important as the vertical. Further details about the used transducer and logger technique can be found in [40]. The horizontal

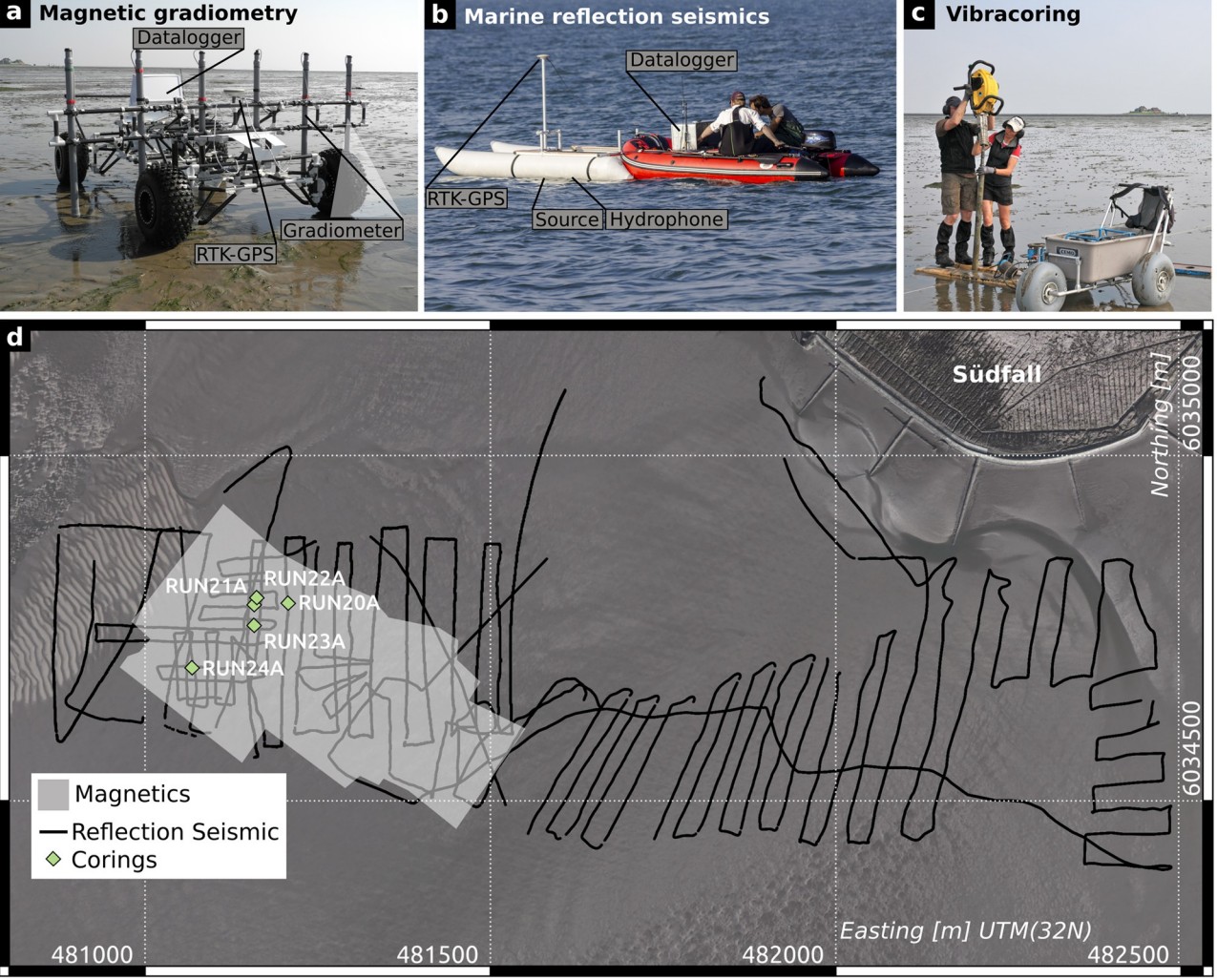

**Fig 3. Methods.** a) Magnetic gradiometer survey cart built for prospection in tidal flat areas. Large wheels prevent the system to get stuck in muddy areas of the tidal flats. b) The used marine seismic acquisition system mounted on an inflatable catamaran in front of a small rubber dinghi. The equipment is made for lightweight transport to the island and based on the system used in [37]. c) Percussion coring in the tidal flat area. Equipment needs to be carried by special carts (seen in the front), d) Map showing the part of the study area that was accessible to magnetics and percussion coring, the position of the corings, and the seismic lines recorded during the two campaigns. Aerial photos are a combination of photos 32480_6034, 32480_6036, 32482_6034, and 32482_6036, republished from Regional government authority for coastal preservation, national park and ocean protection (LKN) under a CC BY license, with permission from LKN, original copyright 2013.

data sampling depends on the signal repetition rate (8 Hz) and travel speed of the boat, which led to an average spacing of about 0.16 m between data points along the boat track. Time data sampling frequency was set to 35.7 kHz with a record length of 57 ms. The positioning is performed by a Leica 1200 RTK-DGNSS (horizontal accuracy of up to 2 cm). Data processing included the following steps:

- Bandpass filtering using a Butterworth filter opening at 1 kHz to 2 kHz and closing from 6 kHz to 7 kHz

- Deconvolution using a fixed filter operator, derived from Wiener predictive error deconvolution of an isolated seafloor reflection signal in deeper water, which is convolved with the full seismic trace.

- Automatic picking of the seafloor reflection and smoothing with a moving average of 120 traces to suppress wave motion, and removal of the seafloor reflection.

- Semblance-based coherence filter [41].

- Geometrical spreading correction using a linear time-gain function.

- Migration of the data using Stolt migration [42] with a constant velocity of 1480 m/s.

Data processing steps were performed based on the software package seismic unix, except the coherence filter which was performed using an own cpp routine added to the seismic unix environment. The profiles of seismic data displayed in Fig 3 were conducted based on the magnetic gradiometry results to get vertical crossections of the magnetic anomalies. Beyond that, seismics allowed to track and extrapolate the observed structures, where magnetic gradiometry cannot go.

## Coring

Sediment cores were drilled at selected locations using an engine-driven coring device (type Atlas Copco Cobra pro) and closed steel augers with plastic liners of 5 cm diameter in order to calibrate the geophysical prospection results. The maximum coring depth reached 4 m below surface (b.s.). All cores were opened, cleaned, photographed, described, and sampled in the laboratory. Descriptions of stratigraphic units followed the standard procedure given by [43, 44] and comprised criteria like grain-size, sediment colour, carbonate content, macrofossil content, archaeological artefacts, etc. Sedimentary logs were created using the GGU-STRATIG software (Civilserve GmbH, Steinfeld, Germany). Analyses of sedimentary, geochemical and microfaunal paleoenvironmental parameters (e.g. grain size, loss on ignition, element concentrations, magnetic susceptibility, foraminifers, and ostracods) allowed to identify different sedimentary environments. For a detailed description of the approach, see [45].

The geochronological framework is based on dendrochronological dating of a wooden beam from the larger tidal gate (published in [7]) and archaeological age estimations of finds from the tidal flats [15].

## Results

### Magnetic gradiometry

Fig 4a shows the magnetic gradient map of the investigated area. In general, the amplitudes of anomalies in the area show a very small dynamic range of about 2 nT. Several features can be observed in the map and are shown in Fig 4b as a redrawing of the visible anomalies. First of all, the map is divided in three main areas (roman numbers I, II, and III), which show a general offset in magnetic signal amplitude in comparison to each other, although single features can be observed continuously passing through all areas. Area I is the area of least magnetic signal amplitude, whereas area III shows a generally increased, irregular signal. Beside these differences, several local features were observed and labeled as follows:

1. Feature IV: an elongated structure passing the entire area.

2. Feature V: several nearly rectangular features that directly connect to the inside of feature IV.

3. Feature VI: elongated structures that intersect feature II

4. Feature VII: two linear anomalies beginning at the position of the tidal gates (based on Fig 2) and going south, then being connected to a faint area of negative anomaly.

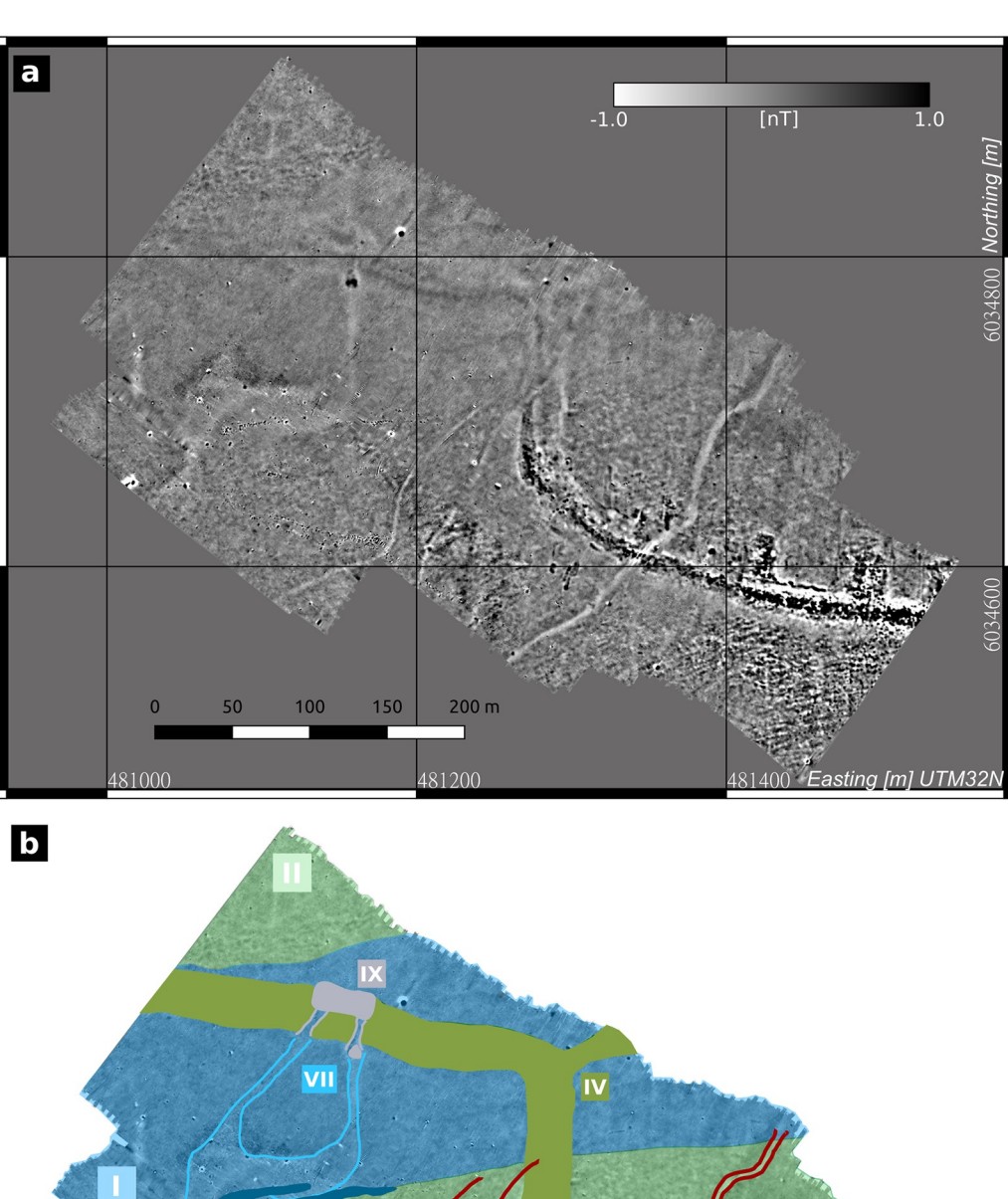

**Fig 4. Results of the magnetic gradiometry.** a) Magnetic gradient map of the tidal gate and western dyke covering the part of the investigation area that was accessible by walking. b) Redrawing of the magnetic map showing the observed main features. Colored areas and lines correspond to roman feature numbers placed in boxes with equal color. I, area of reduced magnetic signal amplitude compared to the areas II and III; IV, elongated structure, related to the dyke; V, signatures that are connected to the inner side of the dyke; VII, anomalies connected to the tidal gate position that is probably related to I; VI elongated anomalies with no corresponding structures in the recorded observations.

5. Feature VIII: elongated accumulations of small dipoles

6. Feature IX: weak anomaly north of the expected positions of the tidal gates.

## Coring

Five cores were taken at representative sites to identify and calibrate the different types of anomalies encountered by magnetic gradiometry and reflection seismics. Regarding the magnetic map, cores RUN 13A (-0.65 m b.s.) and RUN 25A (-0.85 m b.s.) were drilled in areas I and II to the south of the elongate structure of feature IV. RUN 20A (-0.92 m b.s.) and RUN 26A (-0.69 m b.s.) were drilled approx. 200 m apart in feature IV, while RUN 27A (-0.65 m b.s.) lies just north of structure VI at the outer margin of a rectangular feature V. Following [45], seven different facies types were identified from core stratigraphies (Fig 5a). Facies type description follows from bottom to top; labels are according to [45]. Facies type F makes up the lowermost part of all cores and consist of grey to dark grey silty sediments. Deposits reflect low-energy depositional conditions of a brackish to marine shallow water quiet reach environment with predominantly anoxic/reducing conditions sheltered from wave dynamics and/or currents.

Facies type G consist of layers of sand or shell debris that occasionally intersect facies type F. Deposits reflect sporadic increases of wave dynamics and/or currents, e.g. by strong storm surges or shifting tidal inlets.

Facies type E shows light grey silty sediments that are penetrated by thick roots of reed. Deposits reflect a distinct change from brackish-marine conditions of facies type F to

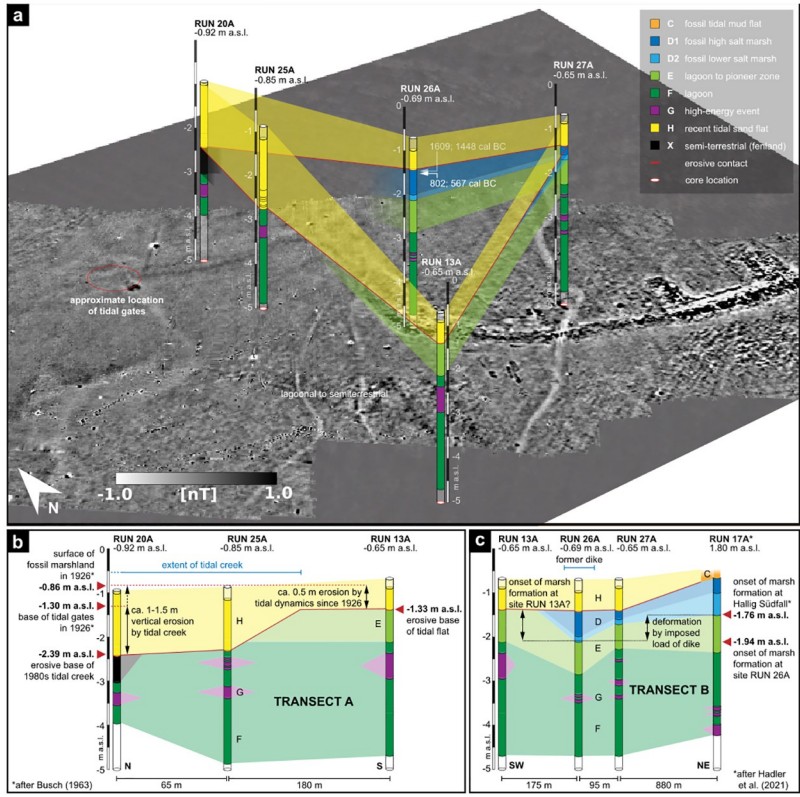

**Fig 5. Results coring transects.** a) Percussion core locations and stratigraphies in relation to magnetic prospection results. b) Transect A shows the increasing thickness of recent tidal flat deposits (facies type H) in northern direction. c) Transect B shows the limited occurrence and different depth levels of facies type D in the eastern part of the study site.

shallow water pioneer zone conditions, characterized by increased salinity (fluctuations) and accumulation of organic-rich mud [46]. Microbial reduction of sea water sulphides is known to significantly enrich Sulphur in freshly deposited muds [47] and a distinct mottling along the upper boundary indicates the development of a characteristic mono- and disulphide zonation [46]. Facies type E was only encountered in cores RUN 13A, RUN 26A and RUN 27A.

Facies type D2 is made of homogeneous mud of light grey colour, indicating a reductive environment. Palaeoenvironmental proxies reflect semi-terrestrial conditions of a frequently flooded pioneer zone to lower salt marsh environment, where constant siltation rises the ground surface to some decimetres below MHW, inducing environmental stress by high fluctuations in salinity and temperature during tides and/or seasons ([46, 48]) as well as initial soil formation processes [49].

Facies type D1 consists of compact, blueish grey clayey mud with strong hydromorphic features. The deposit shows clear characteristics of a high salt marsh environment, where ongoing sedimentation of fine-grained, organic-rich sediments raised the ground level above MHW, intensifying desalinization and soil aeration ([50, 51]). Subsequent oxidation of (Fe-)sulphides and the decay of organic matter produce Fe-oxides, sulphuric and carbonic acid, both causing a rapid decalcification and acidification of the sediment [49]. Covered by younger tidal flat deposits of facies type C, it represents a fossil soil horizon called "Dwogmarsch" ([52–54]).

Facies type D deposits were only found in cores RUN 26A and RUN 27A.

Facies type X consists of compact black organic mud penetrated by thick roots of reed. Sediments likely originate from a (semi-)terrestrial swampy environment. Root canals filled with tidal flat sand indicate a post-depositional exposition in the tidal flats. The unit seems to represent the base of the former dyke that was well visible in the tidal flats in the 1920s and 1930s [8] but became subsequently covered by sediments.

Facies type H finally marks the upper part of all the cores drilled and consists of recent tidal sand flat deposits covering the archaeological remains today. Its lower base is generally marked by a distinct erosive contact.

Core transects A and B (Fig 5a and 5b) highlight both stratigraphic similarities and changes throughout the study area and provide insights to the site's landscape dynamics. Both transects show an even distribution of basal facies type F shallow water quiet reach deposits, locally intersected by sand or shell debris of facies type G. The subsequent facies type E appears in varying depths along transect B but was only discovered to the very south of transect A (RUN 13A). Here, it is covered by ca. 0.5 m of recent tidal flat sediments (facies type H) that become significantly thicker along the northern transect. Facies type H reaches a maximum thickness of ca. 1.5 m at sites RUN 25A and RUN 20A. At site 25A, recent deposits lie directly on top of facies type F (RUN 25A), while at site RUN 20A, they cover the just locally appearing facies type X in a same stratigraphic position. Facies type D occurs only along transect B, extending from Hallig Südfall (RUN 17A, [45]) towards the study area. While the unit is well preserved in the Südfall stratigraphy (RUN 17A), it is significantly thinner at coring site RUN 27A. At site RUN 26A, deposits are again well preserved but lie some decimeters deeper. Unit D is absent at site RUN 13A. Like in transect A, unit H covers the underlying units at all coring sites of transect B with a distinct erosive contact.

## Reflection seismics

As an excerpt of the seismic results, we show example profiles that comprise the basic magnetic features listed above (Fig 6) and then examples on that we were able to perform corings to understand the observed stratigraphy (Fig 7).

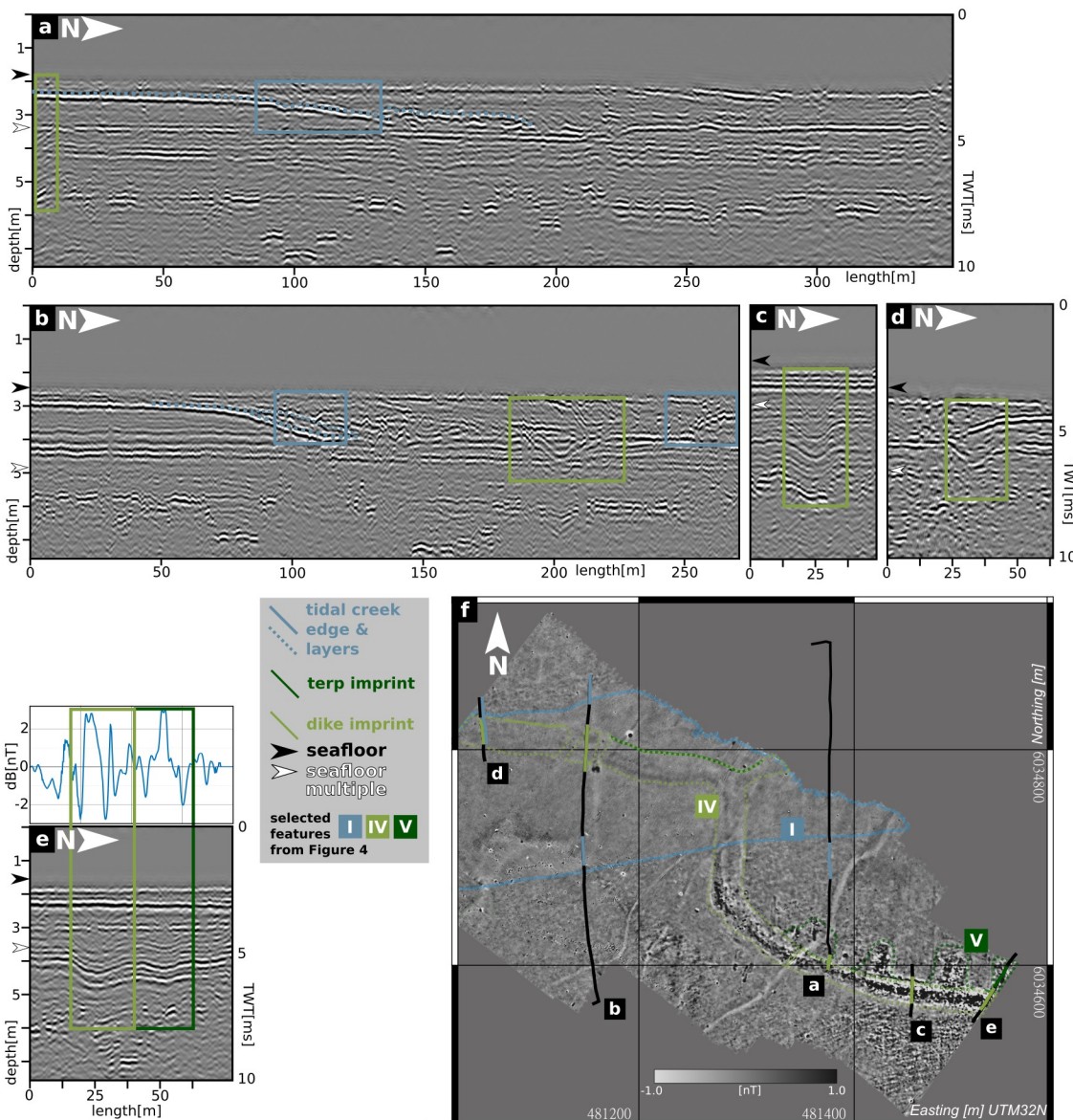

**Fig 6. Results seismic examples.** a)-e) example seismic profiles and a map (f), showing their locations, the magnetic gradient map and the main features highlighted in Fig 4. Colored rectangles (profiles) and lines (map) indicate the position of several features in the magnetic map and their corresponding features in the seismic profiles.

Fig 6 shows five example profiles crossing the magnetic anomalies that were labeled in Fig 4. Fig 6a crosses feature IV and the transition between areas I and II, as well as feature VI. Feature IV is only covered half and is spatially connected to a south dipping reflector at 1 m to 3 m below the seafloor. The transition between I and II corresponds to the beginning of a north dipping reflector in the depth range of 1 m below seafloor. There is no reflection event that corresponds to feature IV. Fig 6b again crosses feature IV and both transitions between I and II. Feature IV now is connected to a depression shaped reflection event starting at about 1 m below seafloor. Also, area I corresponds to a shallow but large depression in the first meter below seafloor. Fig 6c crosses feature IV in a region where it shows higher magnetic signal

amplitude (area II). Again the feature is visible as a sequence of depression shaped reflectors but visible from 1 m to about 4 m below seafloor, whereas when again crossed in area I (Fig 6d), the reflection feature is only visible from 1 m to about 2 m below seafloor. Fig 6e crosses both feature IV and V.

In Fig 7 we combine core stratigraphies with seismic results. Fig 7 lower right shows a map of the seismic profile section and the positions of the corresponding coring sites. Fig 7a offers a complete cross section of the depression shaped reflector in the first meter below the seafloor, which corresponds to the area of weak magnetic amplitude (area I in Fig 4). Core 27A reveals that this top layer can be identified as recent tidal sand flat (facies type H), making the depression and its inner sequence of north dipping reflectors a former tidal creek that was moving

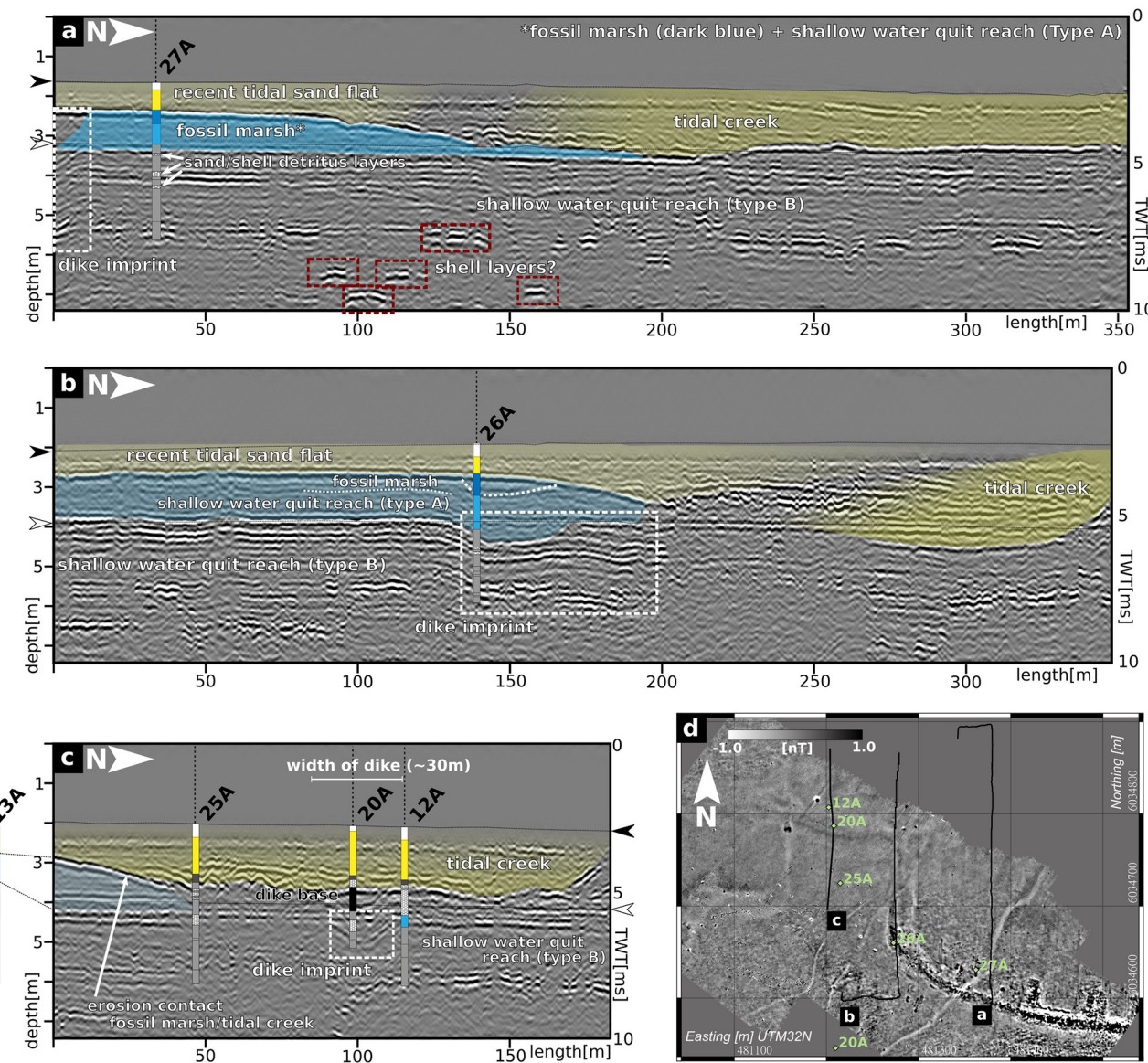

**Fig 7. Seismic and coring results.** a)-c) selected seismic profiles and their preliminary interpretation based on coring results, which are also indicated by coloured bars (facies colors are similar to Fig 5). The positions of the profiles and the cores are shown in the d).

northwards. The same accounts for the top layer of Fig 7b and core 26A. Fig 7a highlights also seismic reflections that correspond to deposits of facies type D and E, thinning towards the center of the tidal creek. Due to the minor thickness of facies type D, there is no visible contrast to facies type E. The northward dipping reflector at the base of the recent tidal flat deposits marks the erosive contact between facies type H and underlying facies types. The reflector's dipping and disappearance of facies types D and E in the northern part of the profile both emphasize the erosion of older sediments by the tidal creek. The lagoonal deposits of facies type E are underlain by similar sediments of facies type F that rather reflect different ecological than deposition conditions, so accordingly the interface alone shows a small impedance contrast in the seismic section. The lagoonal shallow water quiet reach environment of facies type F is intersected by a sequence of shell and sand layers visible as reverberating seismic events. At the deepest part of the profile, high amplitude reflectors at different depths and extents are visible. Fig 7 b) shows a similar stratigraphy except one feature. The uppermost part of the depression shaped reflector at coring site RUN 26A, visible at depths of about 1 m to 4 m and pointed out in Fig 6, corresponds to remains of fossil marsh (facies type D). Unlike in Fig 7a, here facies type D is quite well discernible from underlying facies type E by a depression-shaped reflector. Along the profile, fossil marsh deposits only seem to be preserved within the depression—an observation that corresponds well to the facies type's lower stratigraphic position observed along coring transect B (Fig 5c). In the northern section of the profile, the reflector marking the erosive contact at the base of facies type H reaches its deepest position about 2.5 m below ground surface.

Fig 7c completes the picture of the depression-shaped reflectors. At site RUN 20A, the deeper part of the depression can be assigned to facies type X, associated with the base of the former dyke. The seismic section also shows the extend of erosion by the tidal creek. At coring site RUN 13A, facies type E deposits are still preserved but completely eroded towards the central part of the tidal creek (RUN 25A, RUN 20A). It appears that comparable to site RUN 26A, facies type X deposits are merely preserved as a result of unit deformation and vertical dislocation into the observed depressions. As the slight upward bulge of the erosive contact indicates, facies type X withstood some erosion, likely due to the sediment's compact character.

## Discussion

In this paper, we deal with the prospection of drowned medieval archaeological remains in the tidal flats of the German North Sea coast. Accross the entire North sea area, investigations on paleolandscapes that were drowned due to sealevel rise and storm or Tsunami events are manifold (see e.g. [55] or [56]). A main focus lies on prehistoric sites (see e.g. [57]) like the Mesolithic Doggerland hit by the Storrega tsunami [58]. Prospecting these offshore sites usually involves diving activities in combination with preceding seismic and sidescan or multibeam sonar investigations. Besides these distinct offshore studies, only in Denmark, Germany, the Netherlands, Belgium and Great Britain, noteworthy tidal flat areas that contain archaeological features have been investigated. In Belgium for example, these studies especially involve high resolution marine seismic prospection. [59] for instance demonstrated the potential of 2D/3D seismic echosounder data to image small scale wooden objects in the wadden sea environment. The investigated site belongs to the domain of Walraversijde (Belgium), were the remnants of a late Medieval settlement, both at the beach and inland were investigated. Due to severe coastal erosion, the settlement, dating from the late 13[th] century, was lost to the sea and relocated behind a dyke in the early 15[th] century. This is a different environmental setting compared to the Rungholt settlement, which has been constantly exposed to the wadden sea since the late 13[th] century until today. [60] showed the potential of high resolution seismic

measurements to image the very shallow stratigraphic layers offshore Raversijde, a site with Roman and Medieval remains. They also showed a combination of seismic measurements with terrestrial electromagnetic induction (EMI) as well as Cone Penetration Tests (CPT) at a test area in the intertidal zone, imaging a peat excavation zone. Despite the differences in surroundings and timeframe, these works are strong examples showing the potential of high resolution marine seismic methods to image the very shallow stratigraphy as well as archaeological objects of different scales, including wooden constructions or peat structures in tidal flat areas. In the United Kingdom, besides the investigations of [61] dealing with the Paleolandscape of Doggerland based on exploration seismic and acoustic data, an example of geophysical investigation of a North Sea coastal site is the former medieval port and town of Dunwich which was lost due to cliff erosion and investigated by multibeam-, side-scan sonar-, and singlebeam subbottom seismic data [62]. In the Netherlands, the former tidal flats and salt marshes with remains of settlements from the Pre-Roman Iron Age to the Medieval period are today mostly situated behind the modern dykes, and thus can be investigated by conventional archaeological prospection methods and excavations (see e.g. [63] or [2]). In Germany, different research approaches dealt with the cultural heritage of the wadden sea: In Lower Saxony, several survey campaigns have been performed [64], analysing archives, aerial pictures, archaeological- and geological data. [64] also showed a first result of tidal flat magnetic gradiometry, imaging a farmstead and parts of a tidal creek. In North Frisia, several archaeological surveys were performed in the past (e.g. [6]) as well as a study on synthetic aperture radar (SAR) [65]. [65] showed that high-resolution space-borne SAR imagery with a resolution of one square meter can be used to complement archaeological surveys on intertidal flats. They were able to detect remains of farmhouse foundations and of former systems of ditches, dating to the 13th-17th century AD. Nevertheless, like aerial photography, SAR images are constrained to exposed archaeological features although not limited to daylight. Since SAR-images can be used for large-scale surveillance and provide information on surface roughness, recurrent SAR-prospection might be a tool to monitor ongoing erosion in zones with archaeological remains, as our study demonstrates the scale of losses that can be inflicted by moving tidal creeks. The methodological approach presented in this study offers a very useful solution to settlement prospection in the difficult and demanding environment of tidal-flats in North Frisia and comparable areas. For the first time, a comprehensive picture of the prospected settlement area was gained. The presented Niedam area is a unique case study area, combining both very shallow stratigraphic imaging (including tidal creek erosional interfaces) and the remains of former archaeological structures like dykes and terps as interlinked prospection targets.

To finalize the picture of the presented study area, provided by magnetic gradiometry, reflection seismics and coring, a synoptic interpretation against both the archaeological record and literature is discussed in the following. The basic interpretation of the data starts with the different features in the magnetic map (Fig 4) comparing them to known archaeological structures, observed in the 1920s-1960s (Fig 2). It is striking that feature IV (Fig 4) follows the same path as the observed remains of the medieval dyke. Its exact—and due to the sedimentary cover so far unknown—course can now be precisely located based on magnetic gradiometry. Adding the seismic and coring results, the depression-shaped reflectors holding compact fossil marsh (facies type D in RUN 26A) and organic mud deposits (facies type X, RUN 20A) can also be linked to the medieval dyke system. The unit's local appearance and its stratigraphic position imply that the magnetic results do not show the body but only an imprint of the former dyke, as underlying deposits were deformed by the imposed load of the structure (e.g. the bend-down marsh deposits of facies type D visible in the seismics). The load of the dyke thus leaves an imprint of compressed marshland (facies type D in RUN 26A) or (semi-)terrestrial swampy areas (facies type X in RUN 20A) on which it was built. This effect was already

observed by [66] and summarized by [67] on the exposed remains of the Niedam-dike, but can now be mapped by seismics. This approach provides a valuable tool to track former dyke lines in the Wadden Sea or on land, where all remains of the building itself are gone. The amount of deformation by the dyke may be estimated to at least 20 cm from coring data (RUN 26, Fig 5c), while seismic prospection indicates an even higher amount of deformation of ca. 0.5 m (Fig 7a).

Based on core stratigraphies, we can also derive the cause for changes in magnetic amplitudes between areas I and II. Both seismic and coring results show a thickening and deepening of the recent tidal flat deposits in area I (facies type H in RUN 25A and RUN 20A), as well as sequences of dipping reflection events within this top layer. Compared to maps and aerial images of the study area ([5, 68, 69]), we identified area I as the central channel of a tidal creek active in the 1980s, but silted-up today. The petering of the pre-marsh deposits (facies type E in RUN 13A) towards the center of the tidal creek and a thinner imprint of the dyke (lacking the top layer) attest to severe erosion by the tidal creek in area I. Compared to levellings of the marsh surface exposed in the study area in the 1920s [9], cores and seismic profiles indicate a general lowering of the medieval ground surface of ca. 0.5 m by gradual erosion. In the central part of the tidal inlet, the vertical erosion even reaches 1.5 m to 2 m below present day ground surface (Figs 5b and 7b). These results demonstrate the effects of tidal dynamics (e.g. in form of shifting tidal inlets) to the cultural heritage preserved in the Wadden Sea. With the top layer being eroded and a thick sediment cover on objects in greater depths, the remaining subsurface structures create a much weaker magnetic signal. Our results show that rapidly shifting tidal creeks—a common feature throughout the tidal flats—pose a relevant threat to the cultural heritage, emphasizing the need for proper prospection approaches of the medieval remains preserved in this amphibious landscape.

Besides the tidal creek and the dyke, both yielding the largest anomalies, the magnetic map and seismic profiles also show several smaller features. Regarding the two tidal gates, no evidence of any building structure was discovered in the respective seismic profiles. Wooden beams of the larger tidal gate were still visible and partly recovered in the 1960s by [9], who locates the base of the gate at -1.30 m a.s.l. Reconstructing the depth of the former tidal inlet from seismic profiles and nearby coring sites RUN 20A and RUN 25A (Fig 7c), we conclude that during the last decades, up to 2 m of sediment including the archaeological remains were eroded by the tidal creek and tidal dynamics. It must therefore be assumed, that the last remains of the exceptional tidal gates are finally destroyed and became lost to the sea. There is, however, still evidence of their former location. Just south of the dyke, feature VII in the magnetic map shows two linear anomalies extending south from the gates' assumed position (based on Fig 2) to a basin-like area of slightly negative anomaly. While the former likely reflect channel structures such as drainage ditches, the later strongly resembles some kind of (natural) harbour basin. [8, 9] describe the tidal gates as ca. 25 m long and 40 m apart in an E-W-direction. His measurements fit quite well to the magnetic as well as seismic results, where the imprint of the dyke reveals a basal width of ca. 25-30 m and a distance of ca. 35 m between the anomalies at the gates' assumed locations. We therefore conclude that although the archaeological remains are lost, we were able to reconstruct the former location of the tidal gates and even provide some new evidence for their connection to a harbour site (Fig 8).

Based on our studies, we were able to verify the general observations and locations already recorded at the beginning of the 20[th] century. However, there are considerable deviations from old observations regarding the exact position, but also in terms of interpretation of the individual archaeological features. An overview of the results and a comparison with old recordings is displayed in Fig 8a and 8b. At former times, surface finds were in parts misinterpreted. Besides the position of the dyke and the tidal gates, also the distribution of terps needs to be changed

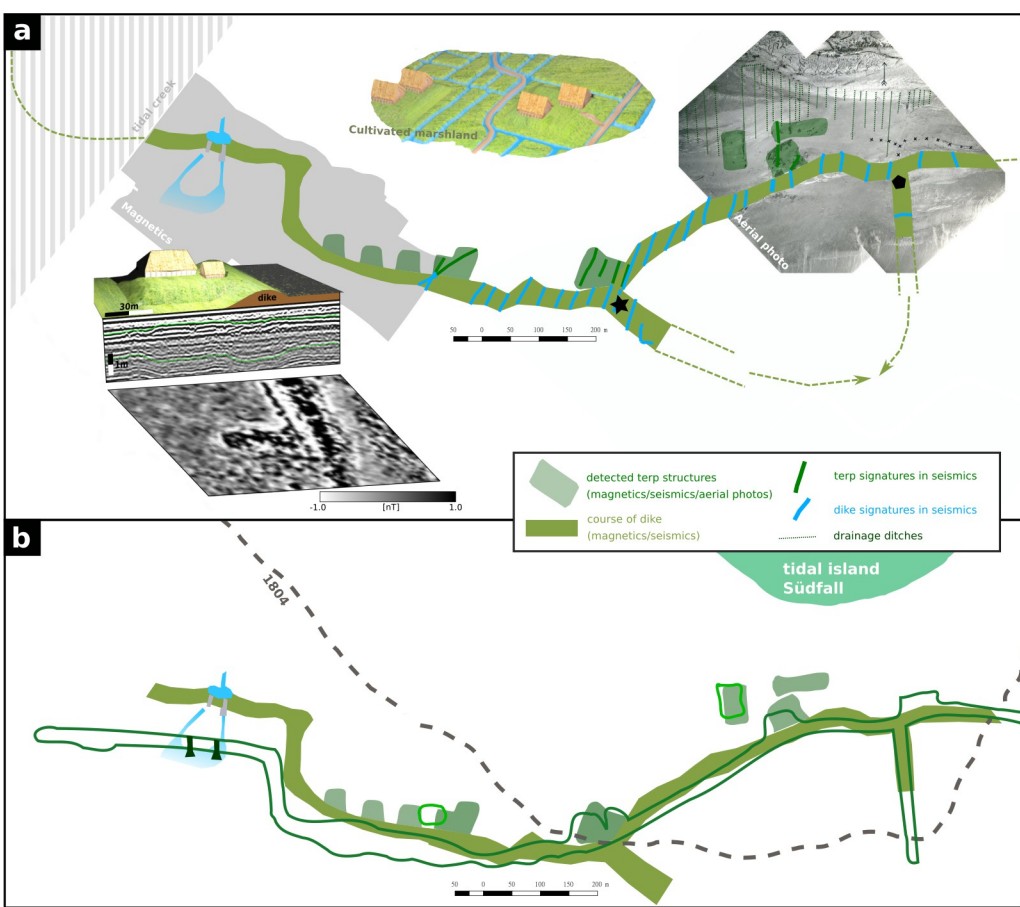

**Fig 8. Interpretation.** a) Final interpretation based on the presented dataset. b) Comparison of the results in a) with the old recordings (lines outlining the features already shown in Fig 2 after [8, 17]).

based on the geophysical prospection results presented by this paper. All terps visible in the magnetic map are connected to the dyke, rectangular in shape and arranged mostly perpendicular to the dyke. They are invariable behind the dyke and not part of the dyke body itself as described by [8, 17]. Furthermore, we did not find a so far presumend large but single terp complex, but instead four rectangular terps located close to each other.

East of this group of terps we could identify a potential dyke branch-off (see position marked by a black star in Fig 8). The same interpretation would apply for another branch-off from the main dyke (marked by a black hexagon in Fig 8), both possibly forming another small polder just south of the Niedam dyke (indicated by the dashed arrow in Fig 8). The polder would fit well to another observation of fossil farmland just south of the main dyke in this area [8].

Comparing the depth of imprints in the seismic results, the imprint of the dyke is deeper than that of the terps associated with it. This would imply that the dyke might have been higher than the terps, questioning the interpretation that only terps were able to protect from the winter storm floods [29]. Furthermore the imprint of the dyke appears symmetrical in its shape, indicating that the former dyke was symmetrical and was not built with a lower seaward slope as expected by [27]. Due to the greater height of the dyke compared to the terps behind it, the exeptionally large width of the dyke base of about 35 m and the installed and once completely

replaced tidal gates, it is likely that the Niedam dyke functioned as a seaward outer dyke. In the literature, however, it is considered to be an inland middle dyke. This is based primarily on the supposed observation that the terps were built on top of the dyke [14, 19, 67]. However, this is not the case according to our prospection results. A comparison with archaeological excavation results of a roughly contemporaneous dyke on the island of Nordstrand also emphasizes the presumed protective function of the Niedam dyke: on Nordstrand, a dyke with a width of about 10 m and a symmetric shape was built remotely from the direct coastline in the 14[th] century, probably as a reaction to the 1362 flood. Only after the loss of its protective function, it was overbuilt with a dyke warft on top ([27]). Especially the great width assigns an active protective function to the Niedam dyke, which is not affected by the terps in its back. Furthermore, the tidal gates and the postulated harbour basin would be nonfunctional, given the dyke was a middle dyke. With this evidence, the prospected part of the Niedam-dyke and its tidal gates can be regarded as a foremost part of Rungholt's seafront and it's maritime infrastructure.

## Conclusion

We present geophysical and geoarchaeological investigations for an area in the North Frisian Wadden Sea holding remains of the sunken medieval settlement of Rungholt. The investigated area is representative for the medieval dyke systems of the region, including harbour structures connected to tidal gates, housing and storage terps. We showed that the delineated prospection setup, using magnetic gradiometry and coring during low tide, and marine reflection seismics during high tide, is highly suitable for imaging and understanding the remains of the former medieval landscape not only in the presented region but also in other parts of the wadden sea. Our results show the critical state of preservation and endangerment of the medieval cultural heritage by the dynamically changing tidal flat environment. Tidal gates, terps and dyke structures, still to be seen in the 1920s, are partly affected by erosion of up to two meters. Based on geophysical and geoarchaeological data, we showed that the imprint of former coastal protection measures on the underlying sediments is still detectable and gives a clear picture of the dyke system, although the associated archaeological remains have already disappeared. Moreover, we detected that terps are largely rectangular and not part of the dyke body itself, but attached to it. We found that these terps were probably not higher than the dyke body making the dyke more important for protection against winter storms. Finally, we showed that the cross-section of the dyke itself was of symmetrical shape and not asymmetrical, as previously thought. The presented work is highly relevant for future geophysical and geoarchaeological prospection in tidal flat areas and a significant base for the reconstruction of passed cultural landscapes. The presented approach enables the investigation of cultural remains and their state of preservation in a landscape of exceptional archaeological value.

## Acknowledgments

The authors would like to thank Detlef Schulte-Kortnack and Clemens Mohr for their extensive technical help, and Gunda and Gonne Erichsen for their logistic support on Hallig Südfall. We would also like to thank the AXIO-NET PED-Service for providing their RTK-correction data.

## Author Contributions

**Conceptualization:** Dennis Wilken, Hanna Hadler, Bente Majchczack, Wolfgang Rabbel.

**Data curation:** Dennis Wilken, Hanna Hadler, Tina Wunderlich, Michaela Schwardt, Annika Fediuk, Peter Fischer, Timo Willershäuser, Stefanie Klooß, Andreas Vött.

**Formal analysis:** Dennis Wilken, Hanna Hadler, Tina Wunderlich, Michaela Schwardt, Annika Fediuk, Peter Fischer, Timo Willershäuser, Stefanie Klooß.

**Funding acquisition:** Dennis Wilken, Hanna Hadler, Tina Wunderlich, Andreas Vött, Wolfgang Rabbel.

**Investigation:** Dennis Wilken, Hanna Hadler, Tina Wunderlich, Michaela Schwardt, Annika Fediuk, Peter Fischer, Timo Willershäuser, Stefanie Klooß, Andreas Vött.

**Methodology:** Dennis Wilken, Hanna Hadler.

**Project administration:** Dennis Wilken, Hanna Hadler, Andreas Vött, Wolfgang Rabbel.

**Resources:** Bente Majchczack.

**Software:** Dennis Wilken, Tina Wunderlich.

**Supervision:** Dennis Wilken, Hanna Hadler, Andreas Vött, Wolfgang Rabbel.

**Validation:** Stefanie Klooß.

**Visualization:** Dennis Wilken, Hanna Hadler.

**Writing – original draft:** Dennis Wilken, Hanna Hadler, Bente Majchczack.

**Writing – review & editing:** Dennis Wilken, Hanna Hadler, Tina Wunderlich, Bente Majchczack, Michaela Schwardt, Peter Fischer, Stefanie Klooß, Andreas Vött, Wolfgang Rabbel.

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
