## [Decision Letter · Decision Letter 0]

12 Apr 2021

PONE-D-21-06483

Lost in the North Sea - Geophysical and geoarchaeological prospection of the Rungholt medieval dyke system (North Frisia, Germany)

PLOS ONE

Dear Dr. Wilken,

Thank you for submitting your manuscript to PLOS ONE. After careful consideration, we feel that it has strong merit but does not fully meet PLOS ONE’s publication criteria as it currently stands. Therefore, we invite you to submit a revised version of the manuscript that addresses the points raised during the review process.

While the reviewers' comments vary, the main points raised, particularly by reviewers 2 and 3, should be taken into account. Particularly, the requested statement on the theoretical context in which the research was performed should be included, as well as broadening the scope of the (geographical) relevance of the research for a wider audience. Equally, parts of the methodology need to be clarified, which is made most explicit by the comments of reviewer 2. To this end, the availability of the borehole data for revision purposes (and as supplementary materials) is needed.Currently, the submitted manuscript reference Hadler et al - submitted) is not accessible, and cannot be used to support the conclusions (see note on data availablity in the publication criteria).

We look forward to receiving your revised manuscript.

Kind regards,

Philippe De Smedt

Academic Editor

PLOS ONE

Journal Requirements:

In your manuscript, please provide additional information regarding the specimens used in your study. Ensure that you have reported specimen numbers and complete repository information, including museum name and geographic location.

For more information on PLOS ONE's requirements for paleontology and archaeology research, see https://journals.plos.org/plosone/s/submission-guidelines#loc-paleontology-and-archaeology-research.

We note that you have stated that you will provide repository information for your data at acceptance. Should your manuscript be accepted for publication, we will hold it until you provide the relevant accession numbers or DOIs necessary to access your data. If you wish to make changes to your Data Availability statement, please describe these changes in your cover letter and we will update your Data Availability statement to reflect the information you provide.

We note that Figures 1, 2, 4, 5, 6 and  in your submission contain map/satellite images which may be copyrighted. All PLOS content is published under the Creative Commons Attribution License (CC BY 4.0), which means that the manuscript, images, and Supporting Information files will be freely available online, and any third party is permitted to access, download, copy, distribute, and use these materials in any way, even commercially, with proper attribution. For these reasons, we cannot publish previously copyrighted maps or satellite images created using proprietary data, such as Google software (Google Maps, Street View, and Earth). For more information, see our copyright guidelines: http://journals.plos.org/plosone/s/licenses-and-copyright.

4a, You may seek permission from the original copyright holder of Figures 1, 2, 4, 5, 6 to publish the content specifically under the CC BY 4.0 license. 

4b, If you are unable to obtain permission from the original copyright holder to publish these figures under the CC BY 4.0 license or if the copyright holder’s requirements are incompatible with the CC BY 4.0 license, please either i) remove the figure or ii) supply a replacement figure that complies with the CC BY 4.0 license. Please check copyright information on all replacement figures and update the figure caption with source information. If applicable, please specify in the figure caption text when a figure is similar but not identical to the original image and is therefore for illustrative purposes only.

We note that Figure 3 includes an image of a participant  in the study.

Reviewers' comments:

Reviewer's Responses to Questions

**Comments to the Author**

1. Is the manuscript technically sound, and do the data support the conclusions?

Reviewer #1: Yes

Reviewer #2: Yes

Reviewer #3: Yes

2. Has the statistical analysis been performed appropriately and rigorously? 

Reviewer #1: Yes

Reviewer #2: N/A

Reviewer #3: N/A

3. Have the authors made all data underlying the findings in their manuscript fully available?

Reviewer #1: Yes

Reviewer #2: No

Reviewer #3: Yes

4. Is the manuscript presented in an intelligible fashion and written in standard English?

Reviewer #1: Yes

Reviewer #2: Yes

Reviewer #3: Yes

5. Review Comments to the Author

Reviewer #1: As always, this contribution from Prof. Rabbel's team is of exceptionally high quality. The work is technically and methodologically innovative, the results are of high quality and of substantial relevance to archaeological as well as near-surface geophysical prospection community & research into the Wadden Sea area.

It has been a pleasure to review this paper.

It becomes clear that this is a multi-author paper since the style of the language changes in different sections.

A section on authorship contributions would be good to attach.

Please put the “st” and “th” attached to all ordinal numerals in superscript: \\textsuperscritp{th} (in the abstract (2x), lines 11, 35, 38 (2x), caption of Fig. 2, 65, 68, 70 (2x), 77, 86, 364, 365, 367, 395 (2x), 471, 504.

Please replace all abbreviations of circa “c.” with “ca.”. You are inconsistent in its use.

Please replace all dashes “-“ with double dashes “--“. It looks much better.

Please remove all spacing between the Figure numbers and sublables (a,b,c): From “Fig. 7 a” to “Fig. 7a”.

When addressing a Figure in the written text, please spell out the word “Figure”.

Please replace all blanks “ “ between numbers and the “m” (metres) with “\\,”, which results in a much nicer, shorter spacing.

In all figure captions, please add a “:” after the bold text.

Please capitalize only the first word in section headings.

Please change the section heading of “History of land use …” to \\section*{History of land use …

In the abstract, second and third sentence, the expression “tidal flats” occurs three times. Please rephrase.

The only question concerning the content I have regarding the statement written in line 66/67. Can you be sure there hasn’t been any earlier, e.g. Mesolithic settlement in the area, like in Doggerland?

In the Email address of the corresponding author a space is missing after “:”.

In affiliation 1, please change “Kiel University” to “Christian-Albrechts …”

Please replace “RTK-DGPS” with “RTK-GNSS”.

In line 239, please spell out “Sulphur”.

Please check the reference “von Carnap”, something odd is going on here.

Please omit the language declaration in the references.

Please be consistent with the inclusion of DOI information in the references.

Please include “\\usepackage{kpfonts}” to render the review experience even more pleasant next time.

In Figure 4b, please add a legend.

Any other typos and issues are highlighted in the manuscript.

Reviewer #2: This paper presents a well-executed case study of archaeological prospection in the intertidal zone using well-established geophysical and geoarchaeological techniques. However, the presentation of the results could be improved. The text and images would benefit from some clarification and finishing. The wider significance of the results for an international audience is currently underreported.

General comments:

-Descriptive results and interpretations are mixed in the results section. The text would be clearer if interpretation or data integration is separated from results of the individual methods (e.g. l278 reference to magnetic gradiometer survey results in coring results)

-Figures in appropriate resolution/size: check if fonts on figures will be readable after resizing

-Finish the figures (horizontal/vertical scale, color/symbol legend, north arrow, coordinates,...). As such they are more easily interpretable on first glance.

Specific comments, ambiguities, questions:

-Abstract in submission system: italics pasted as {\\it Niedam} in text field.

-Data availibilty: Please, provide access link to the data: eg. URL, DOI, WMS, WCS, WFS,... or in supplementary material.

l114: What was the reasoning to select magnetic gradiometry and marine reflection seismic survey instead of other methods?

l116: I assume the magnetic cart is actually a 'non-magnetic cart', a 'magnetometer cart' or a 'magnetic gradiomater survey cart'. I suggest to change the name.

l133: What was the (average) movement speed?

l135: What software was used?

l142: What interpolation method was applied?

l144: Move reference to end of sentence e.g.: following Wilken et al., 2012

l165: What was the spacing between survey lines and why was it chosen? Why not denser? This is partially explaind in results section, but could be moved here.

l167: which data processing software was used?

l180: The applied coring technique is actually not vibracoring but percussion coring, assuming a similar system to this (https://en.eijkelkamp.com/products/augering-soil-sampling-equipment/percussion-drilling-set-gasoline-percussion-hammer.html) was used.

l191: Hadler et al submitted is not accessable (to review) and includes data which are essential to the results (e.g. core descriptions/data). Please, add core data (in appendix) to the paper or refer to a published paper/data.

l198: suggestion: change to 'magnetic map' to 'map of (vertical) magnetic gradient'/'magnetic gradient map'

l199: 'feature' assumes an archaeological/soil feature and an interpretation, which can only be determined after verification/validation. The continued use of 'Anomaly' could be more appropriate in the geophysical results section.

l205: 'Bumpy' is a strange wording. Maybe 'irregular' or 'variable' is more appropriate

l209: Separate data results and interpretation. I suggest to place interpretation either separately in the results section or in the discussion section.

l219-220: better in method section

Figure 5: Fig5a: the perspective image is not an added value and a bit confusing. Better to add core locations on figure 4 and add core lithofacies to figure 5b and 5c. Figure 5b and 5c: include a separate color legend of the lithofacies. Fonts are small.

l227-269: add core description and collected sediment data in a separate table (in appendix). Right now, it is unclear which results were derived from the cores and which are derived from literature.

l268: ...covering the archaeological remains today (today at the end of the sentence).

l278: What explains the difference in magnetic signal?

l289 and l290: move to methodology

Figure 6: suggestion: plot extracted magnetic data profile as a line graph above/on top of the seismic profile to illustrate correlation and mark colored zones of fig 4B. Plot labels on the profiles to mark discussed reflectors and refer to them in the main text. Text discusses depths (m), while two way traveltime (TWT? ms) on figure. Is it possible to label depth? Add horizontal length scale.

l310-311: Figure 7 lower right=> label and refer to as figure 7D. No depth label, no horizontal distance label. Explain color scale or refer to figure 5 for color scale.

Figure 8a: is there a ditch/road parallel to the dike between the terp structures in the magnetic gradiometer data? Fig 8b: add color legend.

l349-402: As presented, this is a literature review and not a discussion of the results. As such, it belongs in introduction, but it is too extensive for this. It would be beneficial to shorten and rewrite this section and integrate it in the actual discussion of the results in or below L402-509.

L511-533: conclusion is written too much as a summary/abstract. Keep the concluding remarks and introduce some perspectives for future research.

L526: atteched=>attached

Reviewer #3: General remarks:

This is a very interesting study on the topic of drowned settlements in the German part of the Wadden Sea region, in which terrestrial and maritime archaeology are connected. I would advise PLOS ONE to accept the paper after some minor revisions. For that, I present an general overview of my comments/suggestions and some additional remarks below.

The authors provide a detailed overview of their technical approach and methodology used to map the remains of the drowned settlement of Rungholt. However, my main concern is that the paper lacks a (short) proper theoretical background. The study region is a truly maritime region and fits well into the concept of the maritime cultural landscape (with phenomena like terps, dikes, salt marshes) as introduced by Christer Westerdahl in 1992 (IJNA, The Maritime Cultural Landscape). Nevertheless, the word ‘maritime’ is not once mentioned in the paper’s main body. Even though PLOS ONE strongly focuses on presenting new technologies and methodologies, a theoretical framework should always be the starting point of a research. For that, I would suggest to add a short paragraph in which the field of research and research scope are introduced and theoretically founded.

Furthermore, I was also wondering why palynological research (and perhaps even macro-botanical research) was not included as part of examining the vibracorings? The presence of specific types of pollen (and seeds) could have been of value for identifying different salt-marsh zones (see e.g. Schepers 2014, Reconstructing Vegetation Diversity in Coastal Landscapes).

Finally, as the results of this study are very promising, I would expect a stronger emphasis on the usefulness of the chosen methodological approach (which would be in line with the scope of PLOS ONE) – not only for Rungholt and North Frisia – but perhaps also for other parts of the Wadden Sea region. This topic is only briefly mentioned in the final part of the Conclusion, whereas – to my concern – it should have been an important topic of the Discussion (which now mainly focuses on the interpretation of the obtained data). In addition, the comparison between methodological research at Rungholt and other parts of the Wadden Sea region (the first part of the Discussion) is perhaps a bit too general: it’s not so much a contribution to the Discussion, but rather a summary.

Some additional minor remarks:

Sentence 9: you might want to mention that the Wadden Sea is a vast coastal region between the northwest of the Netherlands and the south of Denmakr. In general, there are also numerous archaeological remnants in the Wadden Sea region that date back to the Iron Age and Roman era (e.g. Frisian terps of Wijnaldum, Hallum, Hogebeintum and the wierde of e.g. Ezinge). These archaeological features should be considered as the oldest remains of long-term habitation in the Wadden Sea region and should be mentioned in your study. See for example research of Johan Nicolay, Annet Nieuwhof and Albert Egges van Giffen.

Sentence 14: it is true that major storms (re)shaped the North-Frisian coastal region, but this period of ‘heavy weather’ started in the 12th century (see e.g. Duizend jaar wind en weer, by Buisman, 1995).

Sentences 80-85: I couldn’t agree more: this is also the case in the Zuiderzee-region of the Netherlands (an extension of the Wadden Sea); see for example the remains of medieval dikes and terps that surround the former island of Schokland (Van Popta 2020, When the Shore Becomes the Sea).

Sentences 149-150: references should be alphabetically or chronologically organized.

Sentence 167: please specify the accuracy of the RTK-DGPS.

Sentence 218: why did you change the methodology-order of analyses (magnetic gradiometry, marine reflection seismics, vibracoring)?

Sentence 526: ‘atteched’ should be ‘attached’.

Sentence 527: ‘maiking’ should be ‘making’.

6. PLOS authors have the option to publish the peer review history of their article (what does this mean?). If published, this will include your full peer review and any attached files.

Reviewer #1: No

Reviewer #2: No

Reviewer #3: **Yes: **dr. Y.T. (Yftinus) van Popta

---

## [Author Response · Author response to Decision Letter 0]

10 Feb 2022

We would like to thank the editors and reviewers for their effort and constructive comments, which we tried to address in this revised manuscript. Our answers are listed here, as well as in the attached response-pdf

Journal Requirements:

We already used the mentioned template

1. In your manuscript, please provide additional information regarding the specimens used in your study. Ensure that you have reported specimen numbers and complete repository information, including museum name and geographic location.

For more information on PLOS ONE's requirements for paleontology and archaeology research, see https://journals.plos.org/plosone/s/submission-guidelines#loc-paleontology-and-archaeology-research.

No permits were required for the described study, which complied with all relevant regulations.

1. We note that you have stated that you will provide repository information for your data at acceptance. Should your manuscript be accepted for publication, we will hold it until you provide the relevant accession numbers or DOIs necessary to access your data. If you wish to make changes to your Data Availability statement, please describe these changes in your cover letter and we will update your Data Availability statement to reflect the information you provide.

1. We note that Figures 1, 2, 4, 5, 6 and in your submission contain map/satellite images which may be copyrighted. All PLOS content is published under the Creative Commons Attribution License (CC BY 4.0), which means that the manuscript, images, and Supporting Information files will be freely available online, and any third party is permitted to access, download, copy, distribute, and use these materials in any way, even commercially, with proper attribution. For these reasons, we cannot publish previously copyrighted maps or satellite images created using proprietary data, such as Google software (Google Maps, Street View, and Earth). For more information, see our copyright guidelines: http://journals.plos.org/plosone/s/licenses-and-copyright.

We added the appropriate copyright form for all aerial images used in the figures and changed all captions regarding the CCBY4 license. Furthermore alle redrawn maps are now adequately referenced.

 Answer: The content permission form is provided with the resubmission of the Paper

This was already provided based on the PLOSONE permission form.

1. We note that Figure 3 includes an image of a participant in the study.

Answer: Consent forms are added to the submission

We provided the consent forms.

Reviewers' comments:

Reviewer's Responses to Questions

Comments to the Author

1. Is the manuscript technically sound, and do the data support the conclusions?

Reviewer #1: Yes

Reviewer #2: Yes

Reviewer #3: Yes

2. Has the statistical analysis been performed appropriately and rigorously?

Reviewer #1: Yes

Reviewer #2: N/A

Reviewer #3: N/A

3. Have the authors made all data underlying the findings in their manuscript fully available?

Reviewer #1: Yes

Reviewer #2: No

Reviewer #3: Yes

4. Is the manuscript presented in an intelligible fashion and written in standard English?

Reviewer #1: Yes

Reviewer #2: Yes

Reviewer #3: Yes

5. Review Comments to the Author

Reviewer #1: As always, this contribution from Prof. Rabbel's team is of exceptionally high quality. The work is technically and methodologically innovative, the results are of high quality and of substantial relevance to archaeological as well as near-surface geophysical prospection community & research into the Wadden Sea area.

It has been a pleasure to review this paper.

It becomes clear that this is a multi-author paper since the style of the language changes in different sections.

A section on authorship contributions would be good to attach.

Answer: I assumed that PLOS ONE usually provides such information. Otherwise we would add it.

Please put the “st” and “th” attached to all ordinal numerals in superscript: \\textsuperscritp{th} (in the abstract (2x), lines 11, 35, 38 (2x), caption of Fig. 2, 65, 68, 70 (2x), 77, 86, 364, 365, 367, 395 (2x), 471, 504.

Answer: We changed that

Please replace all abbreviations of circa “c.” with “ca.”. You are inconsistent in its use.

Answer: We changed that

Please replace all dashes “-“ with double dashes “--“. It looks much better.

Please remove all spacing between the Figure numbers and sublables (a,b,c): From “Fig. 7 a” to “Fig. 7a”.

Answer: We changed that

When addressing a Figure in the written text, please spell out the word “Figure”.

Answer: We changed that

Please replace all blanks “ “ between numbers and the “m” (metres) with “\\,”, which results in a much nicer, shorter spacing.

Answer: Unfortunately, after several different tries, I couldn’t manage this in latex/overleaf. I probably have missed something.

In all figure captions, please add a “:” after the bold text.

Answer: We added that

Please capitalize only the first word in section headings.

Answer: We changed that

Please change the section heading of “History of land use …” to \\section*{History of land use …

Answer: We changed that

In the abstract, second and third sentence, the expression “tidal flats” occurs three times. Please rephrase.

Answer: We rephrased the sentences

The only question concerning the content I have regarding the statement written in line 66/67. Can you be sure there hasn’t been any earlier, e.g. Mesolithic settlement in the area, like in Doggerland?

Answer: Mesolithic settlements are currently not known from the study area. Mesolithic activity in this area is conceivable, but their archaeological traces would be expected in the deepest layers of the Holocene stratigraphy, due to the mighty sedimentation from the 2nd millenium BC onwards.

We clarified the first known usage of the area in line 66 of the original document.

In the Email address of the corresponding author a space is missing after “:”.

Answer: We changed that

In affiliation 1, please change “Kiel University” to “Christian-Albrechts …”

Answer: We changed that

Please replace “RTK-DGPS” with “RTK-GNSS”.

Answer: We changed that

In line 239, please spell out “Sulphur”.

Answer: We changed that

Please check the reference “von Carnap”, something odd is going on here.

Answer: There was a wrong dash used, probably c.a.p.. We solved this

Please omit the language declaration in the references.

Answer: Based on PLOS-One Author Guidelines, PLOS uses the reference style outlined by the International Committee of Medical Journal Editors (ICMJE), also referred to as the “Vancouver” style.Here, articles not in english should have the language information:

Example: Ellingsen AE, Wilhelmsen I. Sykdomsangst blant medisin- og jusstudenter. Tidsskr Nor Laegeforen. 2002;122(8):785-7. Norwegian.

Please be consistent with the inclusion of DOI information in the references.

Answer: We checked that

Please include “\\usepackage{kpfonts}” to render the review experience even more pleasant next time.

Answer: We will not take that into account

In Figure 4b, please add a legend.

Answer: The legend would only be a roman number (feature) connected to a color. We had the numbers in boxes, corresponding to the area/feature in the figure. We added additional explanation in the caption.

Any other typos and issues are highlighted in the manuscript.

Reviewer #2: This paper presents a well-executed case study of archaeological prospection in the intertidal zone using well-established geophysical and geoarchaeological techniques. However, the presentation of the results could be improved. The text and images would benefit from some clarification and finishing. The wider significance of the results for an international audience is currently underreported.

General comments:

-Descriptive results and interpretations are mixed in the results section. The text would be clearer if interpretation or data integration is separated from results of the individual methods (e.g. l278 reference to magnetic gradiometer survey results in coring results)

Answer: We re-checked the text. Nevertheless, in case of the seismic data, the selection of example profiles needs to be connected to the main features in the gradiometry data, which still is only descriptive, showing what is observed in the seismic data at certain anomalies of the magnetic data.

-Figures in appropriate resolution/size: check if fonts on figures will be readable after resizing

Answer: We checked that and changed parts of the Figures

-Finish the figures (horizontal/vertical scale, color/symbol legend, north arrow, coordinates,...). As such they are more easily interpretable on first glance.

Answer:Scales were provided as scale bars in the original figures. We changed that to axis scales and north arrows. All maps already have coordinates and colorbars. Color symbols in Figure 5 also refer to Figure 7. We added that in the caption.

Specific comments, ambiguities, questions:

-Abstract in submission system: italics pasted as {\\it Niedam} in text field.

Answer: that was simply a format c.a.p. error.

-Data availibilty: Please, provide access link to the data: eg. URL, DOI, WMS, WCS, WFS,... or in supplementary material.

See statement in the submission. DOI will be provided

l114: What was the reasoning to select magnetic gradiometry and marine reflection seismic survey instead of other methods?

Answer: Other available Methods do not provide sufficient penetration (GPR, electromagnetic induction) due to the high salinity of the soil, or sufficient spatial resolution (ERT, land seismic methods, which are also too slow for the narrow time window). We added a paragraph to the Methods section.

l116: I assume the magnetic cart is actually a 'non-magnetic cart', a 'magnetometer cart' or a 'magnetic gradiomater survey cart'. I suggest to change the name.

Answer: We changed that

l133: What was the (average) movement speed?

Answer: The average measurement speed was walking speed, which is difficult to quantify as different persons were using the cart.

l135: What software was used?

Answer: We used our own software package written in matlab. We added that to the text

l142: What interpolation method was applied?

Answer: We used linear interpolation. We added that to the text

l144: Move reference to end of sentence e.g.: following Wilken et al., 2012

Answer: we changed that

l165: What was the spacing between survey lines and why was it chosen? Why not denser? This is partially explained in results section, but could be moved here.

Answer: We moved the sentences from the results section to the end of this paragraph. Profile distance did not follow a certain pattern but in terms of extrapolating the magnetic results, we tried to manage a profile distance of about 30m as best as possible regarding wind and weather conditions and the narrow time frame.

l167: which data processing software was used?

Answer: We used seismic unix and our own c++ tools to process the data. We added that to the text

l180: The applied coring technique is actually not vibracoring but percussion coring, assuming a similar system to this (https://en.eijkelkamp.com/products/augering-soil-sampling-equipment/percussion-drilling-set-gasoline-percussion-hammer.html) was used.

Answer: We changed that in the text. Nevertheless, vibracoring is a common description throughout geoarchaeological literature. 

l191: Hadler et al submitted is not accessible (to review) and includes data which are essential to the results (e.g. core descriptions/data). Please, add core data (in appendix) to the paper or refer to a published paper/data.

Answer: The detailed description of all facies types identified for cores presented within this paper is now published in Hadler et al. 2021, ESPL and therefore be available at the time of publication of this paper. During the revision of this paper, the Hadler et al. 2021 article was accepted for publication. A letter of acceptance is attached.

l198: suggestion: change to 'magnetic map' to 'map of (vertical) magnetic gradient'/'magnetic gradient map'

Answer: we changed to magnetic gradient map.

l199: 'feature' assumes an archaeological/soil feature and an interpretation, which can only be determined after verification/validation. The continued use of 'Anomaly' could be more appropriate in the geophysical results section.

Answer: We do not think that an assumption comes with this word. We think that feature is in general a term that can be used to name both, characteristic anomalies in gradiometry and reflection patterns in seismic. We used ‘feature’ to have a method independent term throughout the paper, that can also describe a group of anomalies. Otherwise ‘anomaly IV’ would depict one single magnetic anomaly, not a family of anomalies.

l205: 'Bumpy' is a strange wording. Maybe 'irregular' or 'variable' is more appropriate

Answer: we changed that to irregular

l209: Separate data results and interpretation. I suggest to place interpretation either separately in the results section or in the discussion section.

Answer: we removed that

l219-220: better in method section

Answer: It refers to the five cores shown here.

Figure 5: Fig5a: the perspective image is not an added value and a bit confusing. Better to add core locations on figure 4 and add core lithofacies to figure 5b and 5c. Figure 5b and 5c: include a separate color legend of the lithofacies. Fonts are small.

Answer: Concerning Fig. 5, we prefer to keep the perspective image of 5a, as it emphasizes - especially in combination with the magnetic map - the observed differences in the magnetic signal, that are caused by (i) an increased thickness of the recent tidal flat sands and (ii) the erosion of the cultural remains by a former tidal creek. The degree and also area of intense erosion become especially clear in the perspective image of the facies distribution and thickness that are later on confirmed and also reflected by the seismic measurements.

However, to clarify the location of each core, we added a symbol to the legend. The latter is now also displayed separately and valid for all parts of the Figure (a-c).

l227-269: add core description and collected sediment data in a separate table (in appendix). Right now, it is unclear which results were derived from the cores and which are derived from literature.

Answer: The data presented in this publication is derived from coring and merely interpreted according to the facies types defined in Hadler et al. (2021). Since a detailed description of each facies type’s characteristics will be given within that paper, we like to refer to that publication for details on coring results. As noted in the text and Fig. 5, the only core derived from literature (which is also Hadler et al. 2021) is RUN 17A. References used in the facies description merely refer to literature that made equal observations on the different proxies.

l268: ...covering the archaeological remains today (today at the end of the sentence).

Answer: we changed that

l278: What explains the difference in magnetic signal?

Answer: The increasing thickness of the sand cover (facies type H) - diamagnetic quartz sand, larger depth of archaeological remains can explain the difference in the magnetic amplitude. Nevertheless, this is part of the discussion.

l289 and l290: move to methodology

Answer: see above

Figure 6: suggestion: plot extracted magnetic data profile as a line graph above/on top of the seismic profile to illustrate correlation and mark colored zones of fig 4B. Plot labels on the profiles to mark discussed reflectors and refer to them in the main text. Text discusses depths (m), while two way traveltime (TWT? ms) on figure. Is it possible to label depth? Add horizontal length scale.

Answer: We added an example magnetic profile for comparison and reworked the figure as proposed.

l310-311: Figure 7 lower right=> label and refer to as figure 7D. No depth label, no horizontal distance label. Explain color scale or refer to figure 5 for color scale.

Answer: Depth labels and horizontal labels were visible as scale bars in the original Figure 6 as well as in Figure 7. We changed that to axis labels. Color scale is referred to Figure 5 in the caption.

Figure 8a: is there a ditch/road parallel to the dike between the terp structures in the magnetic gradiometer data? Fig 8b: add color legend.

Answer: Based on our data, we cannot say whether this is a ditch or road, or simply the edge of the dyke imprint depressen. The legend in 8b is the same as in 8a. We moved it in between both figures.

l349-402: As presented, this is a literature review and not a discussion of the results. As such, it belongs in the introduction, but it is too extensive for this. It would be beneficial to shorten and rewrite this section and integrate it in the actual discussion of the results in or below L402-509.

Answer: To our understanding, a discussion needs to explain what the results mean and why/how they differ from what other researchers have found. One should interpret results in the light of other published results, by adding additional information from sources cited in the introduction section as well as by introducing new sources, which is the main focus of this part of the discussion. It is followed by a more general discussion on methodology. 

L511-533: conclusion is written too much as a summary/abstract. Keep the concluding remarks and introduce some perspectives for future research.

Answer: We moderately reworked the text, trying to highlight concluding points and implications for future work.

L526: atteched=>attached 

Answer: We changed that

Reviewer #3: General remarks:

This is a very interesting study on the topic of drowned settlements in the German part of the Wadden Sea region, in which terrestrial and maritime archaeology are connected. I would advise PLOS ONE to accept the paper after some minor revisions. For that, I present a general overview of my comments/suggestions and some additional remarks below.

1.

The authors provide a detailed overview of their technical approach and methodology used to map the remains of the drowned settlement of Rungholt. However, my main concern is that the paper lacks a (short) proper theoretical background. The study region is a truly maritime region and fits well into the concept of the maritime cultural landscape (with phenomena like terps, dikes, salt marshes) as introduced by Christer Westerdahl in 1992 (IJNA, The Maritime Cultural Landscape). Nevertheless, the word ‘maritime’ is not once mentioned in the paper’s main body. Even though PLOS ONE strongly focuses on presenting new technologies and methodologies, a theoretical framework should always be the starting point of a research. For that, I would suggest to add a short paragraph in which the field of research and research scope are introduced and theoretically founded.

Answer: We agree with the reviewer that the Medieval cultural landscape of North Frisia fits very well into Westerdahl’s concept of a maritime cultural landscape. However, the presented study focuses on a novel geophysical-geoarchaeological approach to investigate only a small section of this landscape. Westerdahl’s holistic considerations of all aspects of the landscape, including natural, cultural and cognitive, reaches far beyond the scope of the presented study. Therefore, we currently see no possibility to address this concept based on available data within this study. An analysis of Medieval North Frisia, based on the theories of Westerdahl, will require a study on it’s own which must be based on archaeological, geophysical and historical data in a much broader geographical extent.

We agree that the term maritime, especially in light of the maritime character of the region, is missing and added it to line 2 of the introduction. Furthermore, we added a sentence after line 509 to emphasize the significance of the revised interpretation of the dyke and terps for the knowledge on Rungholt’s maritime infrastructure.

2.

Furthermore, I was also wondering why palynological research (and perhaps even macro-botanical research) was not included as part of examining the vibracorings? The presence of specific types of pollen (and seeds) could have been of value for identifying different salt-marsh zones (see e.g. Schepers 2014, Reconstructing Vegetation Diversity in Coastal Landscapes).

Answer: Palynological analyses were not included in the research, since the authors focused on microfaunal analysis that reflected the fossil marshes also quite well. However, preservation conditions for pollen should be quite good throughout the study area and are planned for future core analyses in the second phase of the project.

3. 

Finally, as the results of this study are very promising, I would expect a stronger emphasis on the usefulness of the chosen methodological approach (which would be in line with the scope of PLOS ONE) – not only for Rungholt and North Frisia – but perhaps also for other parts of the Wadden Sea region. This topic is only briefly mentioned in the final part of the Conclusion, whereas – to my concern – it should have been an important topic of the Discussion (which now mainly focuses on the interpretation of the obtained data). In addition, the comparison between methodological research at Rungholt and other parts of the Wadden Sea region (the first part of the Discussion) is perhaps a bit too general: it’s not so much a contribution to the Discussion, but rather a summary.

Answer: in terms of the conclusion we fully agree and tried to highlight this aspect. The discussion has the parts, one setting the results into an archaeological context, by discussing their contribution to the understanding of finds as described in the literature, the second part discusses the presented methodology with respect to other approaches and their environments. The third discusses the general usefulness of the approach and its benefit in wadden regions. We tried to highlight that.

Some additional minor remarks:

Sentence 9: you might want to mention that the Wadden Sea is a vast coastal region between the northwest of the Netherlands and the south of Denmakr. In general, there are also numerous archaeological remnants in the Wadden Sea region that date back to the Iron Age and Roman era (e.g. Frisian terps of Wijnaldum, Hallum, Hogebeintum and the wierde of e.g. Ezinge). These archaeological features should be considered as the oldest remains of long-term habitation in the Wadden Sea region and should be mentioned in your study. See for example research of Johan Nicolay, Annet Nieuwhof and Albert Egges van Giffen.

Answer: We included the remark for sentence 9

We agree that the first permanent settlements from the Iron Ages are somewhat lacking in the text. Nordfriesland differs in this point from the Netherlands, Lower Saxony and Dithmarschen, since a widespread settlement of marshes does not occur in the Iron Age or Roman Iron Age. Settlement finds of these Periods are quite scarce. We clarified this at two points in the text: In the introduction we added a sentence about the onset of permanent settlement in the general Wadden Sea region from 600 BC (after Line 2) with regard to the Dutch research tradition (Bazelmans et al. 2012). With regard to the North Frisian situation, we added a remark in line 67.

Sentence 14: it is true that major storms (re)shaped the North-Frisian coastal region, but this period of ‘heavy weather’ started in the 12th century (see e.g. Duizend jaar wind en weer, by Buisman, 1995).

Answer: In the case of North Frisia, the historic records for medieval storm events are thin and often contradictory. Although there are mentions of floods in the late 12th and 13th century (e.g. summarized by Jensen and Müller-Navarra, Storm Surges on the German Coast, Die Küste 74, 2008 or by Sear, Southern East Coast North Sea Storms Database, 2018), we have no secure evidence how North Frisia was affected. It is only clear from the historical and archaeological sources, that the coastline retreated strongly during the 14th century with the 1362-event as a major factor. 

We clarified this by a revision in line 17.

Sentences 80-85: I couldn’t agree more: this is also the case in the Zuiderzee-region of the Netherlands (an extension of the Wadden Sea); see for example the remains of medieval dikes and terps that surround the former island of Schokland (Van Popta 2020, When the Shore Becomes the Sea).

Sentences 149-150: references should be alphabetically or chronologically organized.

Answer: we changed that to chronologically

Sentence 167: please specify the accuracy of the RTK-DGPS.

Answer: we added that

Sentence 218: why did you change the methodology-order of analyses (magnetic gradiometry, marine reflection seismics, vibracoring)?

Answer: We changed the order because we are referring to coring results in the seismic data, plotting both reflection seismic sections and corings in Figure 7.

Sentence 526: ‘atteched’ should be ‘attached’.

Answer: We changed that

Sentence 527: ‘maiking’ should be ‘making’.

Answer: We changed that

---

## [Decision Letter · Decision Letter 1]

3 Mar 2022

Lost in the North Sea - Geophysical and geoarchaeological prospection of the Rungholt medieval dyke system (North Frisia, Germany)

PONE-D-21-06483R1

Dear Dr. Wilken,

We’re pleased to inform you that your manuscript has been judged scientifically suitable for publication and will be formally accepted for publication once it meets all outstanding technical requirements.

Kind regards,

Philippe De Smedt

Academic Editor

PLOS ONE

Additional Editor Comments (optional):

Reviewers' comments:

Reviewer's Responses to Questions

**Comments to the Author**

1. If the authors have adequately addressed your comments raised in a previous round of review and you feel that this manuscript is now acceptable for publication, you may indicate that here to bypass the “Comments to the Author” section, enter your conflict of interest statement in the “Confidential to Editor” section, and submit your "Accept" recommendation.

Reviewer #2: All comments have been addressed

2. Is the manuscript technically sound, and do the data support the conclusions?

Reviewer #2: Yes

3. Has the statistical analysis been performed appropriately and rigorously? 

Reviewer #2: Yes

4. Have the authors made all data underlying the findings in their manuscript fully available?

Reviewer #2: Yes

5. Is the manuscript presented in an intelligible fashion and written in standard English?

Reviewer #2: Yes

6. Review Comments to the Author

Reviewer #2: All comments have been addressed. In my opinion, the paper is now acceptable for publication.

Congratulations on the impressive results.

7. PLOS authors have the option to publish the peer review history of their article (what does this mean?). If published, this will include your full peer review and any attached files.

Reviewer #2: No

---

## [Editor Report · Acceptance letter]

25 Mar 2022

PONE-D-21-06483R1 

Lost in the North Sea - Geophysical and geoarchaeological prospection of the Rungholt medieval dyke system (North Frisia, Germany) 

Dear Dr. Wilken:

I'm pleased to inform you that your manuscript has been deemed suitable for publication in PLOS ONE. Congratulations! Your manuscript is now with our production department. 

Kind regards, 

on behalf of

Dr. Philippe De Smedt 

Academic Editor

PLOS ONE